# DreamCatcher: A Wearer-aware Sleep Event Dataset Based on Earables in Non-restrictive Environments

**Zeyu Wang**[1*], **Xiyuxing Zhang**[1*], **Ruotong Yu**[1*]

Yuntao Wang[1†], Kenneth Christofferson[2], Jingru Zhang[1], Alex Mariakakis[2],
Yuanchun Shi[1,3]

[1] Department of Computer Science and Technology, Tsinghua University[‡]

[2] University of Toronto     [3] Qinghai University

`wang-zy23@mails.tsinghua.edu.cn`, `yuntaowang@tsinghua.edu.cn`

## Abstract

Poor quality sleep can be characterized by the occurrence of events ranging from body movement to breathing impairment. Widely available earbuds equipped with sensors (also known as earables) can be combined with a sleep event detection algorithm to offer a convenient alternative to laborious clinical tests for individuals suffering from sleep disorders. Although various solutions utilizing such devices have been proposed to detect sleep events, they ignore the fact that individuals often share sleeping spaces with roommates or couples. To address this issue, we introduce DreamCatcher, the first publicly available dataset for wearer-aware sleep event algorithm development on earables. DreamCatcher encompasses eight distinct sleep events, including synchronous dual-channel audio and motion data collected from 12 pairs (24 participants) totaling 210 hours (420 hour.person) with fine-grained label. We tested multiple benchmark models on three tasks related to sleep event detection, demonstrating the usability and unique challenge of DreamCatcher. We hope that the proposed Dream-Catcher can inspire other researchers to further explore efficient wearer-aware human vocal activity sensing on earables. DreamCatcher is publicly available at `https://github.com/thuhci/DreamCatcher`.

## 1 Introduction

More than one-seventh of the global population suffers from at least one kind of sleep disorder, yet many are undiagnosed [6, 36, 41]. Sleep disorders can lead to various health issues, such as cardiovascular disease and depression [14, 20, 39]. The gold-standard diagnostic method, polysomnography (PSG), requires patients to spend the night in a specialized sleep clinic. Conducting such sleep studies can be cost-prohibitive and resource-intensive. Additionally, patients may suffer from the "first-night effect" where they exhibit anomalous sleep behavior when spending the night in a new environment [29]. These challenges call for a minimally intrusive at-home sleep monitoring solution that can alert wearers to potential sleep disorders.

---

[*]equal contribution

[†]corresponding author

[‡]Key Laboratory of Pervasive Computing, Ministry of Education, Beijing National Research Center for Information Science and Technology, Department of Computer Science and Technology, Tsinghua University, Beijing, 100084, China

38th Conference on Neural Information Processing Systems (NeurIPS 2024) Track on Datasets and Benchmarks.

Many sleep disorders are associated with at least one detectable sleep event. For instance, obstructive sleep apnea (OSA) is characterized by the sudden cessation of snoring [4, 30], bruxism manifests as frequent teeth grinding or clenching [19], and restless sleep is often accompanied by excessive nighttime movement [41]. Because it is difficult for people to recall these events while sleeping, continuous monitoring is crucial to facilitate diagnosis.

Recent research has shown that lightweight earables [8] can provide convenient real-time monitoring of human activity [11, 34, 45, 52, 46]. For sleep monitoring in particular, earables have unique advantages over other wearables like smartwatches and smartphones [3, 9, 24, 25, 48]. The ears are located on the head and close to the trunk of the body, allowing microphones to capture rich acoustic information generated during sleep. The in-ear feedback microphone included in active noise-cancelling earbuds can even detect subtle sounds produced within the body. For this work, we utilize a modified commercial earbud containing two microphones (feedback and feedfoward) and an inertial measurement unit (IMU).

Advancements in hardware technology and machine learning algorithms have spurred increased research into sleep monitoring using commodity wearables. Current acoustic-based sleep event detection algorithms mainly focus on audio feature engineering [2, 13] or lightweight deep learning models [11]. These solutions are often developed using data collected in controlled environments and contrived scenarios with minimal confounds (e.g., ambient noise). However, people often share sleep spaces with other individuals like roommates or spouses who may move and create sounds, leading to observable events not associated with the wearer [9, 13]. Moreover, these studies have not made their code or datasets publicly available.

We address these shortcomings by presenting and releasing DreamCatcher — a large-scale, multi-modal, multi-sleeper sleep event dataset of earable data with fine-grained labels. We recruited 12 pairs (24 participants) of people who slept in the same room, and one person from each pair had a potential sleep disorder. We collected earable data from these pairs over the course of 420 hours and manually annotated 8 sleep events: teeth grinding, swallowing, somniloquy, breathing, coughing, snoring, and body movement. To demonstrate the utility of DreamCatcher, we present case studies of how it can be used to train baseline models that address three valuable tasks: wearer event identification, wearer-aware sleep sound event classification, and wearer-aware sleep sound event detection.

The main contributions of this work are as follows:

- We collected and released the first and largest sleep dataset based on multi-modal earable data collected in real scenarios with the disruption of sleep partners. Data is synchronized and annotated with fine-grained event labels.

- We benchmarked DreamCatcher on three sleep monitoring tasks: wearer event identification, wearer-aware sleep sound event classification, and wearer-aware sleep sound event detection.

- We provide open-source resources including the dataset, code for setting up benchmarks, and tutorial for constructing the earable hardware we used.

## 2 Related Work

### 2.1 Contactless Sleep Monitoring and Wearer-Awareness

The gold standard for sleep monitoring is polysomnography (PSG), which entails wiring a series of sensors onto an individual in a sleep clinic for continuous monitoring and observation. PSG sessions are expensive, labour-intensive, and time-consuming [28], so researchers have shown substantial interest in developing more convenient sleep monitoring solutions suitable for home use.

Contactless sleep monitoring typically falls under one of two methods. The first method involves acoustic sensing of audible sounds using smartphones [24], smartwatches [9, 12], and earbuds [2, 11, 13, 33]. While these systems can work with commodity devices, they are prone to interference in multi-user settings. The second method relies on wireless sensing to detect body motion, respiration, and even heartbeats through minor chest movements at specific frequencies. Commonly used signals include WiFi [50], mmWave [49], and sonar [27]. Wireless sensing makes it possible to manage multi-user scenarios since reflections from multiple users arrive at different times [49, 27]. However, dedicated devices for such approaches are non-trivial to deploy, and wireless sensing is less effective for detecting sleep events such as snoring, swallowing, or somniloquy.

In summary, acoustic and wireless solutions to contactless sleep monitoring show promise in addressing the multi-user challenge, yet cater to different aspects of sleep monitoring. Moreover, as indicated in [15], there is an inherent trade-off between accuracy and comfort.

## 2.2 Sleep Monitoring with Earables

Compared to other commodity wearables for sleep monitoring, earables are worn in closer proximity to respiratory-vocal system and the external carotid artery, offering an ideal position for measuring behaviors and physiological parameters related to sleep [35]. These opportunities have been leveraged using specialized biomedical sensors for sleep monitoring around the ear, such as in-ear EEG [23, 26, 40] and PPG [44] sensors, but these sensors are not widely available on commercial earables due to their high cost and integration complexity.

Some recent works have explored sleep monitoring by leveraging earables without modification, relying on motion sensors and in-ear microphones used for active noise cancellation. Leveraging the audio signals from earables, Ren et al. [33] developed a system that could track breathing rate and detect four sleep events. Christofferson et al. [11] utilized microphones in commercial earbuds for sleep sound classification. Their proposed SleepTSM model achieved promising performance in detecting seven different sleep events with a small footprint suitable for deployment on earables. Han et al. [13] proposed EarSleep, a similar sleep stage classification system dependent on acoustic sensing of body sounds.

Although the microphones on commercial earables have been used to great effect in sleep monitoring, such systems are often evaluated in controlled scenarios with a single participant at a time. In multi-sleeper scenarios, sounds may originate from people who are not wearing the earable, leading to mischaracterizations of the wearer's sleep experiences. Drawing inspiration from the EarSAVAS dataset [51], our work on DreamCatcher facilitates the development and evaluation of wearer-aware ubiquitous acoustic sleep event monitoring systems by providing a public sleep dataset that encompasses not only the wearer's sound events but also interference from non-wearers and non-restrictive environmental conditions.

## 2.3 Sleep Datasets

Table 1 compares datasets across sensing modalities that have been leveraged for sleep monitoring research. It reveals multiple data-related challenges faced by previous works:

| Method | Device | Modalities | | Scale | | Scenario | | Open-source |
| --- | --- | --- | --- | --- | --- | --- | --- | --- |
| | | Acoustic | Non-Acoustic | Data Amount | Participants | Real | Open | |
| SleepEDF [17] | PSG | ✗ | ✓ | 197 nights | – | ✓ | ✗ | ✓ |
| MASS [31] | PSG | ✗ | ✓ | 200 nights | 200 | ✓ | ✗ | ✓ |
| SleepHunter [12] | smartphone | ✓ | ✓ | 90 nights | 45 | ✓ | ✗ | ✗ |
| SleepGuard [9] | smartwatch | ✓ | ✓ | 210 nights | 15 | ✓ | ✗ | ✗ |
| FusedTSNet [2] | – | ✓ | ✗ | 1 hour | – | ✗ | ✗ | ✗ |
| SleepTSM [11] | earable | ✓ | ✗ | 6 hours | 20 | ✗ | ✗ | ✗ |
| EarSleep [13] | earable | ✓ | ✗ | 48 nights | 18 | ✓ | ✗ | ✗ |
| Ren et al. [33] | earable/smartphone | ✓ | ✗ | – | 6 | ✓ | ✗ | ✗ |
| DreamCatcher (ours) | earable | ✓ (2-channel) | ✓ | 420 hours / 62 nights | 24 | ✓ | ✓ | ✓ |

Real: whether events were real or simulated; Open: whether multiple individuals were in the same room

Table 1: Sleep Study Dataset Comparison.

1. Although there has been substantial work utilizing ubiquitous sensors such as microphones in commodity wearables, the only publicly available datasets are for gold-standard PSG.

2. Previous wearable solutions reliant on audio phenomena have overlooked the interference of non-wearer in multiple sleeper scenarios.

3. Proprietary datasets using commodity wearables are typically limited in scale, both in terms of the number of participants and the quantity of data per person. This issue is particularly prevalent in research related to earables [34].

Our dataset aims to fill the gaps. To the best of our knowledge, DreamCatcher is the first open-source sleep event dataset targeted at ubiquitous sensors on commercial devices. Previous work has only considered single-sleeper scenarios. By integrating data from non-wearers, DreamCatcher facilitates

the development and evaluation of wearer-aware sleep event monitoring. Our dataset consists of synchronous dual-channel audio and motion data collected from 12 pairs (24 participants) totaling 210 hours (420 hour.person) with fine-grained labels of eight distinct sleep events. As the largest open-source sleep dataset to date, we envision DreamCatcher will advance wearer-aware sleep event monitoring on commercial earables.

## 3 DreamCatcher Dataset

### 3.1 Dataset Collection

**Hardware.** Using a commodity earable is important because custom devices can often be optimized for data quality in ways that do not translate to existing platforms. Because commodity earables do not provide API access to their data streams, we had to modify an earbud for data acquisition. As shown in Figure 1a, we integrated an MPU6050 IMU sensor into the hardware of Bose QC 20 earbuds, preserving the native feedback and feedforward microphone configuration. All sensors were controlled by a compact external development board, wherein the audio signal was sampled at 24 kHz and the IMU signal was sampled at approximately 94 Hz. To enhance user comfort, the development board was integrated into an enclosure and wrapped like a necklace. This device can function continuously for roughly 7 hours. Appendix A.1 contains more implementation details.

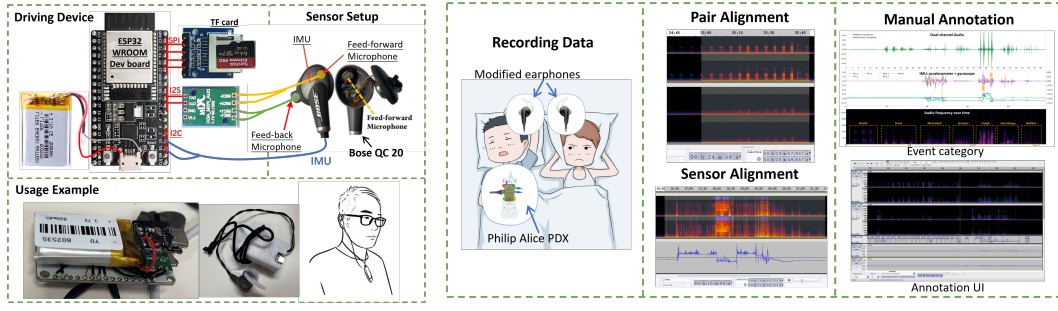

(a) Hardware.  (b) Multi-sleeper Data Collection.

Figure 1: Experiment Setup.

For gold-standard sleep monitoring data, we used a portable PSG system by Philips called the Alice PDx [4]. This device includes a canula and thermal sensor for measuring airflow in the nose, a chest strap for measuring chest expansion during breathing, a fingertip SpO2 sensor for measuring peripheral oxygen saturation, and limb movement sensors. This device was only worn by the person in each participant pair who reported a sleep disorder.

**Participants.** To collect data while accounting for the influence of a sleep partner, we recruited participants in dyads, imposing no constraints on their relationship as long as they shared a bedroom. A total of 12 pairs (24 participants) participated in the study, including 9 males and 15 females aged between 19 and 51 (average = 24.7, standard deviation = 8.3). Among the 12 participants who were equipped with the PDx device, 6 individuals were observed to exhibit varying degrees of sleep apnea.

**Experiment Protocol.** As shown in the first step of Figure 1b, each pair of participants slept in the same bedroom for at least 6 hours. They were instructed to start their sleep session around the same time to maximize the amount of temporal overlap in their data. They were also asked to set an alarm clock that could be heard by both earphones; this sound was used for post-hoc manual data alignment. After the participants woke up and turned off the data collection hardware, they were required to fill out a PSQI [7] questionnaire to self-report their sleep quality.

---

[4] https://www.usa.philips.com/healthcare/product/HC1043844/
alice-pdx-portable-sleep-diagnostic-system

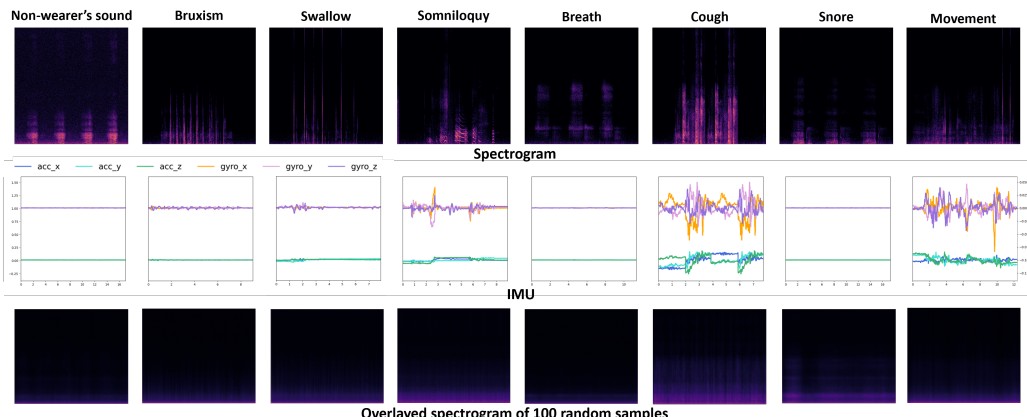

Figure 2: Examples of Each Sleep Event.

## 3.2 Annotation and Statistics

**Data Alignment.** As shown in the second step of Figure 1b, we performed post-hoc alignment for (1) the audio and IMU data in each earable and (2) each pair's audio data. The first round of alignment involving the data modalities within each earbud would hypothetically be trivial because the sensors should be intrinsically synchronized as they are connected to the same ESP32 board and controlled by the same microcontroller. However, we observed an inherent clock drift between the audio and IMU sampling protocols. Over a span of 7 hours, the IMU recording extended 3 seconds longer than the audio, accounting for a deviation of about 0.01%. To correct this, we re-scale the IMU data to match the audio recording duration, as the drift is evenly distributed over the entire recording period.

Because each participant's data was recorded independently by separate earbuds, the second round of alignment involved aligning data across participant pairs. To accomplish this, we utilized an alarm clock as a compensatory reference, manually adjusting the audio recordings to align with the alarm clock's spectrogram.

**Annotation.** Because data was collected from participants' homes, using video was not an acceptable form of annotation due to privacy concerns. Instead, we set up a hierarchical inspection process in which a team of annotators reviewed the earbud data to identify and label events. The annotators were asked to identify the eight sleep events listed in Table 2. They used Audacity[5] to inspect each participant's binaural audio channel and IMU data as well as the sleep partner's binaural audio data simultaneously. The IMU data helped annotators determine the category of wearer-emitted events, while the sleep partner's audio helped them determine whether the event was emitted by the wearer. The annotation process, described more thoroughly in Appendix A.4, entailed selecting an interval for each event they noticed and then assigning a category to it. Each label was checked by at least three annotators; whenever they did not reach a consensus, voting was used to assign labels. Examples of each event are provided in Figure 2.

**Dataset Statistics.** Table 2 summarizes the prevalence and duration of each event type, while Figure 3a illustrates the distribution of durations of each event type. DreamCatcher is a highly imbalanced dataset, reflecting the natural scarcity of certain sleep disturbances such as bruxism, swallowing, somniloquy, and coughing.

Figure 3b shows the smoothed average frequency of different events over the course of a typical night of sleep. Note that participants slept for different amounts of time, so there may be some misalignment in the timing of events across individuals. However, the plot reflects some known observations about sleeping. For example, movement and swallowing were less prevalent after the first hour of sleep, while snoring and somniloquy became more prevalent.

---

[5]https://www.audacityteam.org/

| Label | Description | Total Duration (hrs) | Avg Duration (secs) | S.D. (secs) |
|---|---|---|---|---|
| Noise | Acoustic events emitted by non-wearers, as well as background noises | 32.78 | 2.27 | 2.49 |
| Bruxism | Grinding or clenching teeth | 5.15 | 2.00 | 4.01 |
| Swallow | Reflexively or intentionally saliva swallowing | 1.28 | 1.75 | 1.89 |
| Somniloquy | Talking aloud, murmuring, or shouting while asleep | 0.37 | 1.49 | 2.39 |
| Breathe | One inhalation + one exhalation | 83.56 | 2.21 | 2.41 |
| Cough | Coughing, throat clearing, or sniffling | 0.04 | 1.60 | 1.24 |
| Snore | One inhalation + one exhalation with prominent vibrations or whistling | 31.98 | 3.10 | 3.97 |
| Movement | Shifts in position or gestures made | 10.51 | 6.83 | 7.17 |

Table 2: Label Definitions and Summary Statistics.

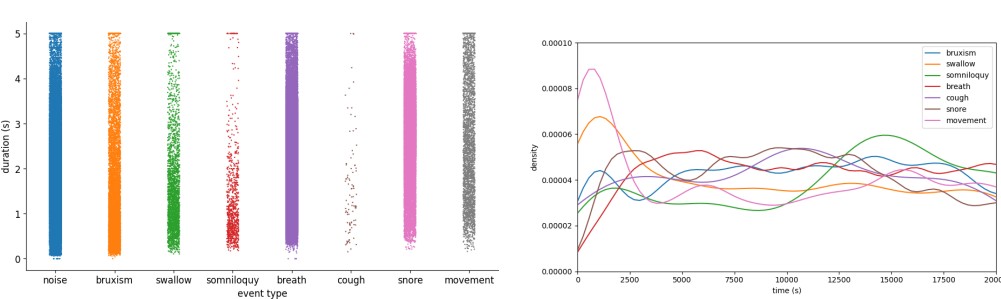

(a) Label duration distribution for each category.    (b) Averaged label density overnight.

Figure 3: Label Distributions.

**Participant Data Splits.** To evaluate how a sleep monitoring system would generalize to unseen users without any calibration data, we recommend splitting data according to participant IDs. Each participant in our dataset exhibited different distributions of sleep events; this information is detailed in Appendix A.4 A.2. The split configurations that optimize the balance in label prevalence are depicted in Figure 4.

### 3.3 Ethics and Accessibility

The protocol used to generate the DreamCatcher dataset received approval from the local Institutional Review Board (IRB) where the data was collected. Participants were explicitly informed about the data recording process and that the dataset would be made publicly available. To safeguard participants' privacy, DreamCatcher has been fully anonymized. An important consideration in this regard is the fact that participants recorded data at home for an entire night, so any private conversations may have been recorded by the earables' microphones. To address this concern, all non-somniloquy dialogue was manually removed from the dataset before it was released.

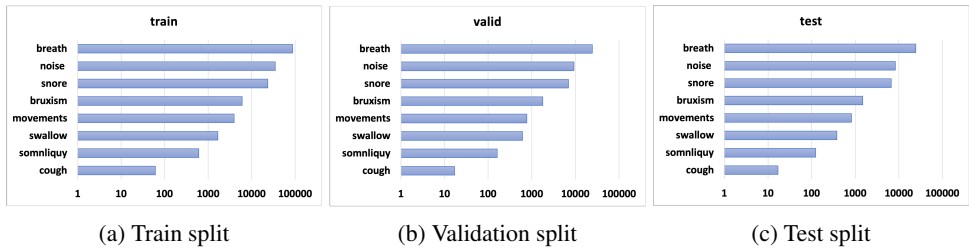

(a) Train split    (b) Validation split    (c) Test split

Figure 4: Cross-user Splits.

# 4 Benchmarks

## 4.1 Wearer Event Identification

**Task Description.** In multi-sleeper scenarios, sleep event monitoring using acoustic methods suffers from the interference of external sounds not produced by the earable wearer. Therefore, we define wearer identification as a binary classification task focused on determining whether audio events come from the wearer or other sources. To the best of our knowledge, no previous work has explored this topic in the context of sleep event monitoring.

**Dataset Preparation.** For this task, we assume that candidate events have been separated from silence using a simple threshold-based approach and focus only on data that our annotators labeled. After segmenting synchronous dual-channel audio and motion data according to the fine-grained labels, we extracted the features listed in Table 3a from each event. The primary challenge for this task is the design of features that are computationally efficient for wearable devices with limited processing capabilities, capable of distinguishing events caused by the sleep partner.

Due to the low intensity of background noise typically seen in real-world sleep scenarios, we relied on acoustic features to distinguish between wearer and non-wearer events. We calculated traditional acoustic features like zero-crossing rate (ZCR) and root mean square (RMS) on both the feedforward and feedback channels. We also calculated three inter-channel audio features — RMS-ED, Mel-FD, and TDOA — that model the different propagation characteristics of sounds. Based on the observation that bone-conducted sound from the wearer should have higher energy at the feedback microphone than at the feedforward microphone, RMS-ED measures the root mean square of the energy difference between the two audio channels. Given the different propagation paths of wearer and non-wearer sounds reaching the ear, Mel-FD measures the energy difference between the Fast Fourier Transforms (FFTs) calculated from both audio channels according to the Mel scale. Finally, time difference of arrival (TDOA) between the two channels reflects the propagation path difference between wearer and non-wearer sounds.

Since the wearer's sleep events are often accompanied by body movements that are captured by the earables' motion sensors, we also extracted motion-related features. We first calculated the overall magnitude of the accelerometer and gyroscope data separately, after which we computed IMU-STD as the mean and standard deviation of those magnitudes over time.

**Benchmark Methods.** Given the low dimensionality of the input data, we benchmarked five traditional machine learning models: two low-complexity models (logistic regression and linear SVM) and three high-complexity models (random forest, decision tree, and AdaBoost).

**Model Training and Evaluation Metrics.** To evaluate the performance of the models, we trained and evaluated each one using leave-one-user-out cross-validation. Since this is a binary classification task, we report performance according to accuracy, F1 score, precision, recall, and AUC.

**Results.** Table 3a shows that most of our features could be computed using fewer than 1 M FLOPs across our entire dataset. The results in Table 3b demonstrate that the high-complexity models achieved similarly higher accuracy compared to the low-complexity ones. According to the feature importance scores presented in Appendix B.1, we found that models with higher complexity are more effective in leveraging the inter-channel audio features. Furthermore, we observed that motion features were important for all models, particularly those that were less complex. These results highlight the utility of inter-channel audio and motion features for wearer awareness of sleep monitoring.

|  | Modality | FLOPs (M) |
|---|---|---|
| ZCR | per-channel (audio) | 0.4 |
| RMS | per-channel (audio) | 0.4 |
| RMS-ED | inter-channel (audio) | 0.6 |
| Mel-FD | inter-channel (audio) | 6.1 |
| TDOA | inter-channel (audio) | 2.7 |
| IMU-STD | per-sensor (accel & gyro) | 0.5 |

(a) Comparison of Extracted Features.

| Method | Acc. | AUC | Recall | Prec. | F1 |
|---|---|---|---|---|---|
| Random Forest | **0.997** | **0.999** | **0.998** | **0.998** | **0.998** |
| Decision Tree | 0.993 | 0.990 | 0.996 | 0.996 | 0.996 |
| AdaBoost | 0.922 | 0.965 | 0.960 | 0.947 | 0.951 |
| Logistic Regression | 0.797 | 0.597 | 0.994 | 0.800 | 0.886 |
| SVM (linear) | 0.602 | 0.539 | 0.645 | 0.816 | 0.721 |

(b) Comparison of ML Algorithms.

Table 3: Benchmarks for Wearer Event Identification.

| Method | input channel | Evaluation Metrics (%) | | | | FLOPs (G) | Params. (M) |
|---|---|---|---|---|---|---|---|
| | | Acc. | Macro-AUC | Macro-F1 | MCC | | |
| SleepTSM [11] | feedback | 73.01 | 63.51 | 35.61 | 49.98 | 0.927 | 0.37 |
| SleepTSM | feedforward | 72.89 | 64.00 | 36.97 | 50.76 | 0.927 | 0.37 |
| EarVAS [51] | all | 78.07 | 74.49 | 36.76 | 49.22 | 0.354 | 12.90 |
| EarVAS | dual-channel-audio | 76.64 | 77.36 | 38.77 | 51.12 | 0.040 | 4.40 |
| EarVAS | feedback | 75.75 | 75.75 | 34.68 | 46.84 | 0.040 | 4.40 |
| EarVAS | feedforward | 76.99 | 75.06 | 34.76 | 43.02 | 0.040 | 4.40 |
| EarVAS | imu-only | 45.03 | 60.56 | 13.75 | 12.24 | 0.313 | 8.50 |
| BEATs [10] | feedback | 90.73 | 80.38 | 57.47 | 66.60 | 22.46 | 90.51 |
| BEATs | feedforward | 89.64 | 78.93 | 55.51 | 59.04 | 22.46 | 90.51 |
| Wav2Vec2.0 [5] | feedback | 75.45 | 91.60 | 48.36 | 56.72 | 26.84 | 94.39 |
| Wav2Vec2.0 | feedforward | 73.29 | 88.84 | 42.52 | 54.11 | 26.84 | 94.39 |
| CLAP (zero-shot) [47] | feedback | 37.04 | 65.57 | 16.77 | 18.21 | 53.03 | 190.80 |
| CLAP (zero-shot) | feedforward | 37.35 | 65.64 | 17.31 | 18.98 | 53.03 | 190.80 |

Table 4: Benchmarks for Wearer-Aware Sleep Sound Event Classification.

## 4.2 Wearer-Aware Sleep Sound Event Classification

**Task Description.** Sleep sound event classification serves as the foundation of sleep disorder diagnosis. Han et al. [13] also revealed that the categorization of sleep sound events also facilitates sleep stage inference. Although algorithms already exist for this task, the interference caused by non-wearers is often overlooked, limiting their applicability in multi-sleeper scenarios. Inspired by EarSAVAS [51], we define wearer-aware sleep sound event classification as an $(n + 1)$-class multi-classification task, where $n$ represents the number of target events and the remaining class encompasses both ambient and non-wearer sounds.

**Dataset Preparation.** As with wearer event identification, we assume that event onset and offset are already known for this task. To standardize the input data size, we cropped the synchronous audio and motion data into 5-second clips that were sufficiently long to cover the duration of the longest event in our dataset.

**Benchmark Methods.** We examined five state-of-the-art models for this task:

1. **SleepTSM** [11] is a lightweight sleep sound classification model that was not evaluated with multiple sleepers in the same room.

2. **EarVAS [51] and its variants** were evaluated on the EarSAVAS dataset to demonstrate subject-aware vocal activity classification utilizing dual-channel audio and motion data.

3. **Wav2Vec2.0** [5], **BEATs** [10], and **CLAP** [47] are generic audio event classification methods.

Besides EarVAS, the other models are only designed to support single-channel audio input. Since DreamCatcher includes audio from both the feedforward and feedback microphones, we evaluated these models on each of those channels separately. All model pre-processing steps and hyperparameters were configured identically to those in the original works we replicated. Appendix C.1 shows the details of the partition of our dataset and the training details of every benchmark model.

**Model Training and Evaluation Metrics.** Each model was evaluated using leave-one-user-out cross-validation. Since this is a multi-class task, we used accuracy, macro-averaged AUC, macro-averaged F1 score, and MCC as evaluation metrics. We also report model complexity according to FLOPs and the number of parameters.

**Results.** As shown in Table 4, most of the models achieved accuracies above 70%; the exceptions were CLAP and a configuration of EarVAS that only used IMU data. However, the macro-F1 scores were typically far lower. This is largely due to the significant class imbalance of our dataset, as some events are far more common than others. Another hurdle encountered by these models was the challenge of jointly optimizing wearer event identification and sleep sound classification. We used a single model to perform both tasks simultaneously, but a dual-stage pipeline may be more appropriate in future work. Appendix C.2 provides a more thorough analysis of the results, showing the efficacy of the feedback microphone in detecting low-intensity events like swallowing and highlighting the promise of sensor fusion for future explorations.

| Method | input channel | Evaluation Metrics | | FLOPs (G) | Params. (M) |
| | | Macro-F1 (%) | Error Rate | | |
| --- | --- | --- | --- | --- | --- |
| SEDNet [1] | dual-channel audio | 14.02 | 0.97 | 0.31 | 0.37 |
| SEDNet | feedback | 18.85 | 0.91 | 0.30 | 0.37 |
| SEDNet | feedforward | 17.98 | 0.98 | 0.30 | 0.37 |
| ATST-SED [38] | feedback | 24.73 | 0.85 | 44.16 | 172.9 |
| ATST-SED | feedforward | 24.10 | 0.85 | 44.16 | 172.9 |

Table 5: Benchmarks for Wearer-Aware Sleep Sound Event Detection.

### 4.3 Wearer-Aware Sleep Sound Event Detection

**Task Description.** Knowing when a sleep event starts and stops is crucial for sleep monitoring, as the temporal distribution and order of events provide critical insights into sleep progression [13]. Inspired by sound event detection (SED) systems and the DCASE Challenge [18], we define wearer-aware sleep sound event detection as a task that involves determining not only the category of an event but also its onset and offset.

**Dataset Preparation.** Following the data format standards from the DCASE Challenges [18], we used 10-second clips for this experiment so that the models would have enough context for precise event detection.

**Benchmark Methods.** State-of-the-art sound event detection methods predominantly employ deep learning, with most of them being built upon convolutional recurrent neural networks (CRNNs). According to Mesaros et al. [22], such methods have been both trained from scratch and have utilized transfer learning to shortcut learning. We benchmarked SEDNet [1] and ATST-SED [38] to represent these two categories, respectively. We selected SEDNet because of its pioneering role in using CRNNs with multi-channel microphone data for sound event detection. On the other hand, we selected ATST-SED because it outperformed all competitors on the DESED dataset [42].

**Model Training and Evaluation Metrics.** Each model was evaluated using leave-one-user-out cross-validation. We used conventional collar-based metrics [21] including event-based macro-averaged F1 score and error rate to quantify model performance.

**Results.** According to the results shown in Table 5, we found that ATST-SED achieved significantly better performance at the cost of a much larger footprint. We also observed that both models were more accurate when they were trained using feedforward microphone audio. In fact, SEDNet trained on multiple audio channels achieved the lowest macro-F1 score out of all the configurations we tested. Appendix D.3 provides a more thorough analysis of the results.

## 5 Limitations and Future Work

First of all, the natural frequency distribution of sleep events leads to a highly imbalanced dataset in DreamCatcher, with rarer events often holding greater significance. Based on the DreamCatcher dataset, we generated a balanced dataset through data augmentation methods and trained classification models on it, as shown in Appendix C.3. We envision the generation of rare sleep events, which must aligns with the patterns of human sleep, will be a highly valuable area for future research.

Moreover, privacy concerns in multi-sleeper settings preclude video verification, resulting in potential label inaccuracies despite requiring a consensus among at least three annotators for challenging-to-identify events. Furthermore, the emergence of commercial earphones equipped with physiological sensors like photoplethysmography (PPG) presents an opportunity to enhance DreamCatcher with additional data modalities in future iterations.

The current prototype earbuds used in our study may not be the epitome of comfort for all users, especially for those who have difficulties falling asleep. However, the existence of commercially available sleep earbuds that are small, soft, and ergonomically designed (e.g., Amazfit Zenbuds[6] and

---

[6]https://www.amazfit.com/products/amazfit-zenbuds

Bose Sleepbuds[7]) underscores the potential for earbuds to become a comfortable and viable sleep monitoring platform. These options point towards a promising future for the application of earbud technology in sleep studies.

## 6 Conclusion

This paper introduces DreamCatcher, the first open-source dataset featuring multi-sleeper, multi-modal data from a commodity device along with fine-grained annotations of sleep disorder-related sound events. DreamCatcher encompasses 420 hours of synchronized dual-channel audio and motion data, offering a rich and challenging resource for sleep monitoring. We validated DreamCatcher's utility by establishing benchmarks across three distinct tasks, and we hope that these results motivate other researchers to innovate further on our dataset.

## 7 Acknowledgement

This work is supported by Natural Science Foundation of China under Grant No. 62472244, No. 62132010 and No. 62222606, University of Toronto – Tsinghua University Joint Research Fund, Natural Sciences and Engineering Research Council of Canada (NSERC) Discovery Grant (#RGPIN-2021-03457), Tsinghua University Initiative Scientistic Research Program, Beijing Key Lab of Networked Multimedia, Institute for Artificial Intelligence, Tsinghua University (THUAI). Thank to the participants involved in data collection and to the professional annotators who contributed to data labeling.

---

[7]`https://www.bose.ca/en/p/all-health/bose-noisemasking-sleepbuds/SBD-SLEEPBUDS.html`

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

# A  Experiment Detail

## A.1  Hardware

The detailed hardware implementation is described in Table 6. Temperature was recorded but not used during annotation and benchmarking models.

| Module | Hardware | Frequency | Configuration Detail |
|---|---|---|---|
| dual-microphones | Bose QC 20 | 24 kHz | I2S protocol, 16-bit PDM data format |
| IMU | MPU6090 | ≈94 Hz | I2C protocol, 7-channel including accelerometer, gyroscope and thermometer |
| compute chip | ESP32-WROOM | 240 MHz | data transfer at 100KB/s with SPI protocol and an external TF card, 3.7V 800 mAH battery |

Table 6: Hardware Implementation Detail.

## A.2  Anonymized Participant Information

We recruited participants for our study through a campus study recruitment platform that included students, faculty, and their family members. Table 7 shows the self-reported sleep disorder information for all participants along with whether any apnea events were detected by those who wore the PDX device.

| Pair # | P1 Reported Disorder | P1 Apnea Detected (Y/N) | P2 Reported Disorder |
|---|---|---|---|
| 1 | Snore/Bruxism/Somniloquy | Y | — |
| 2 | Snore | Y | – |
| 3 | Bruxism | N | – |
| 4 | Snore/Bruxism | Y | – |
| 5 | – | N | – |
| 6 | Snore | N | – |
| 7 | Snore/Bruxism | N | Snore |
| 8 | Snore | Y | Snore |
| 9 | Bruxism/Somniloquy | N | – |
| 10 | Bruxism/Somniloquy | N | Somniloquy |
| 11 | Snore | Y | – |

Table 7: Self-reported Sleep and Detected Sleep Disorders Among Participants.

## A.3  Protocol and Compensation

To familiarize study participants with the data collection hardware and protocol, they were given a video tutorial along with the following set of instructions:

1. Wear the PDx device according to the user guide and video to ensure that the sensors are working properly.

2. Put on the headphones and start recording. To synchronize the data, please set an alarm on a networked mobile phone for the nearest whole hour (e.g., 9 PM, 10 PM) after the headphones begin recording. When the alarm sounds, please announce the time loud enough so that it can be heard by both sets of headphones.

3. You should wear the headphones throughout the night for three consecutive nights, ensuring at least 6 hours of data recording each night.

4. You should charge the devices after each recording so that they are ready for the next night's experiment.

5. After waking up each morning, please fill out a sleep diary to indicate when you fell asleep, when you woke up, and any instances when you woke up during the night. In addition, please fill out a PSQI sleep quality questionnaire.

Given that the hourly minimum wage was $10 USD where this research was conducted, participants were paid $70 USD per night of sleep. We collected 62 nights of data in total, totaling $4,340 USD across the entire study.

### A.4 Manual Annotation

Figure 5 shows the Audacity UI annotators used to examine and label data. The interface included each participant's dual-channel audio and six-channel IMU data. Annotators created a separate "annotation" track where they could set the start and stop times of different events along with their labels. Table 8 shows the total number of events identified within each participant's data.

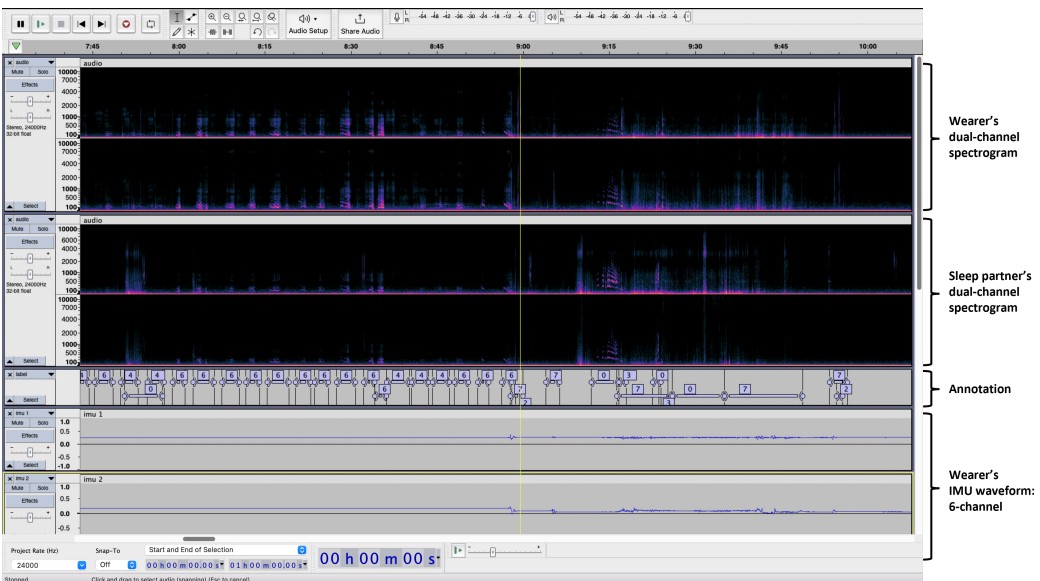

Figure 5: Annotation Software UI.

| User | Label | | | | | | | |
|------|-------------------|---------|---------|------------|---------|-------|-------|----------|
|      | Non-wearer's Sounds | Bruxism | Swallow | Somniloquy | Breathe | Cough | Snore | Movement |
| 1-1  | 634   | 450  | 121 | 68  | 4565   | 32 | 13396 | 320 |
| 1-2  | 16720 | 399  | 214 | 86  | 2580   | 4  | 96    | 393 |
| 2-1  | 734   | 588  | 197 | 61  | 9629   | 6  | 3153  | 325 |
| 2-2  | 2400  | 118  | 116 | 100 | 3682   | 2  | 4     | 596 |
| 3-1  | 708   | 499  | 167 | 13  | 5684   | 2  | 328   | 266 |
| 3-2  | 882   | 245  | 90  | 10  | 1308   | 2  | 12    | 293 |
| 4-1  | 1059  | 478  | 319 | 6   | 8447   | 14 | 2419  | 217 |
| 4-2  | 5061  | 53   | 21  | 1   | 831    | 7  | 19    | 153 |
| 5-1  | 523   | 407  | 222 | 27  | 5589   | 1  | 18    | 438 |
| 5-2  | 614   | 599  | 131 | 26  | 9491   | 3  | 153   | 311 |
| 6-1  | 206   | 1050 | 102 | 36  | 12210  | 2  | 470   | 316 |
| 6-2  | 202   | 428  | 151 | 12  | 10248  | 3  | 484   | 226 |
| 7-1  | 7547  | 655  | 178 | 17  | 8280   | 3  | 2778  | 184 |
| 7-2  | 2600  | 240  | 183 | 36  | 8512   | 4  | 6126  | 246 |
| 8-1  | 43    | 81   | 26  | 6   | 2602   | 1  | 2245  | 47  |
| 8-2  | 2161  | 199  | 19  | 4   | 63     | 1  | 31    | 47  |
| 9-1  | 267   | 661  | 57  | 71  | 6954   | 3  | 328   | 148 |
| 9-2  | 2197  | 321  | 39  | 75  | 5708   | 0  | 11    | 177 |
| 10-1 | 316   | 519  | 85  | 93  | 8617   | 2  | 205   | 195 |
| 10-2 | 464   | 633  | 97  | 129 | 6160   | 0  | 26    | 320 |
| 11-1 | 1     | 43   | 43  | 6   | 578    | 0  | 1739  | 40  |
| 11-2 | 1847  | 37   | 10  | 1   | 1060   | 0  | 4     | 31  |
| 12-1 | 1058  | 658  | 62  | 30  | 9802   | 3  | 3178  | 162 |
| 12-2 | 4615  | 126  | 50  | 1   | 5071   | 0  | 7     | 139 |
| **Total** | **52859** | **9487** | **2700** | **915** | **137671** | **95** | **37230** | **5590** |

Table 8: Count of Sleep Events Per Participant.

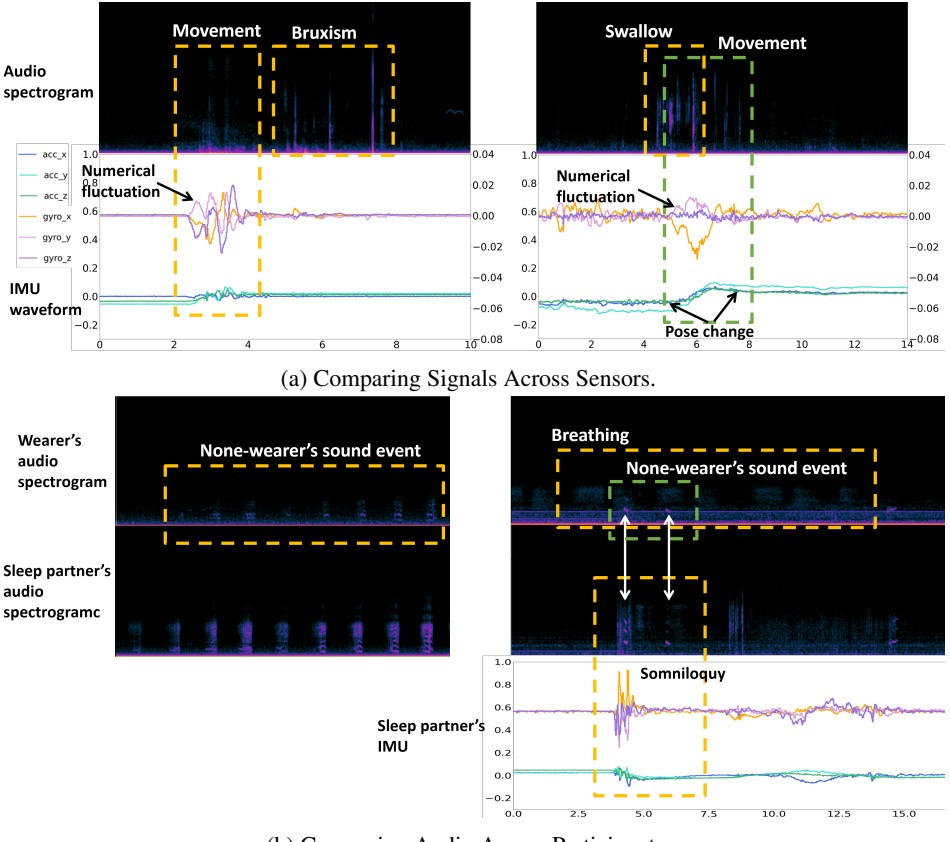

(a) Comparing Signals Across Sensors.

(b) Comparing Audio Across Participants.

Figure 6: Aligned Data Annotation.

As shown in Figure 6, comparing data across wearers and non-wearers helped annotators identify the sources of different sounds. While loud breathing, snoring, or body movement by one individual can be captured by the earbuds of another in the same room, annotators were able to attribute the event's origin to the device that recorded the higher audio intensity. In some cases, the movement data also facilitated annotation. For example, the audio spectrograms of swallowing and body movement are quite similar and often occur simultaneously. However, the latter typically induces large fluctuations in IMU data due to changes in posture while swallowing is more subtle.

## B Benchmark: Wearer Event Identification

### B.1 Supplementary Results

Figure 7 shows the importance of the audio and motion features we used in the lightweight and complex models for this task.

## C Benchmark: Wearer-Aware Sleep Sound Event Classification

### C.1 Model Architectures

All of the models were built using PyTorch 1.13.0 and trained on an NVIDIA GeForce RTX 4090 GPU. We followed each paper as closely as possible and leveraged the accompanying code when available, using 5-second-long data inputs. The following describes the specific implementations we employed in this benchmark:

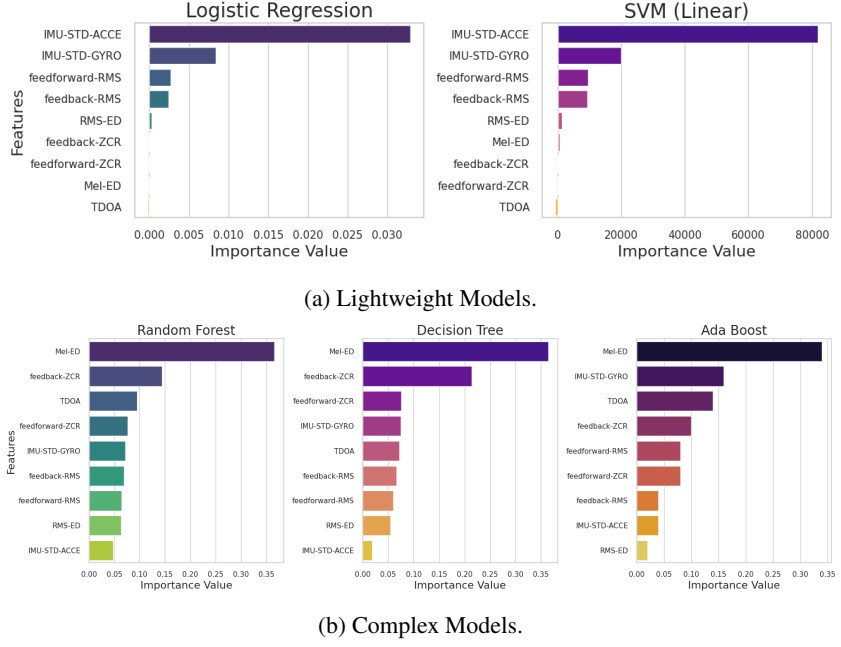

(a) Lightweight Models.

(b) Complex Models.

Figure 7: Feature Importances for the Wearer Event Identification Models.

**SleepTSM [11]:** We processed audio segments into log-scaled Mel filter bank features using a 0.1-s Hanning window with a 2-ms stride, yielding a $128 \times 1501$ input for each channel. Since SleepTSM is not open-sourced, we re-implemented it as described in the paper. The model was trained for 50 epochs using the Adam optimizer with a constant learning rate of 1e-4, a batch size of $32 \times 4$, and a warmup ratio of 0.1.

**EarVAS [51]:** We processed audio segments into log-scaled Mel filter bank features using a 25-ms Hanning window with a 10-ms stride, yielding a $2 \times 498 \times 128$ input across both channels. As a requirement of the model's EfficientNet-B0 sub-module, we implemented zero-padding on the filter-bank features along the time axis to obtain $2 \times 512 \times 128$ (channel×time×frequency) feature vectors. The motion input of EarVAS is raw 6-axis IMU data without any pre-processing. For the model itself, we leveraged the open-source code available at `https://github.com/thuhci/EarSAVAS`, which is shared under the MIT License found at `https://github.com/thuhci/EarSAVAS?tab=MIT-1-ov-file`. The model used SpecAugment [32] for data augmentation and was trained for 30 epochs using the Adam optimizer with a constant learning rate of 1e-4 and a batch size of 128.

**Wav2Vec2.0 [5]:** We processed audio segments using the authors' bespoke feature extractor. We then finetuned their open-sourced base model found at `https://huggingface.co/facebook/wav2vec2-base`, which is shared under the Apache v2.0 License found at `https://huggingface.co/datasets/choosealicense/licenses/blob/main/markdown/apache-2.0.md`. The model was trained for 10 epochs using the Adam optimizer with a constant learning rate of 3e-5, a batch size of $32 \times 4$, and a warmup ratio of 0.1.

**BEATs [10]:** We processed audio segments into log-scaled Mel filter bank features using a 25-ms Hanning window with a 10-ms stride, yielding $498 \times 128$ input for each channel. We leveraged the open-source model available at `https://github.com/thuhci/EarSAVAS/tree/main/BEATs_on_EarSAVAS`, which is licensed under the MIT License found at `https://github.com/thuhci/EarSAVAS?tab=MIT-1-ov-file`. The model was trained for 30 epochs using the Adam optimizer with a constant learning rate of 1e-4 and a batch size of 32.

**CLAP [47]:** We performed zero-shot classification on the open-source model available at `https://huggingface.co/laion/larger_clap_general` under the Apache v2.0

## C.2 Supplementary Results

Figure 8 shows the confusion matrices associated with all of the models that were trained in this benchmark. The rest of this subsection describes the notable trends.

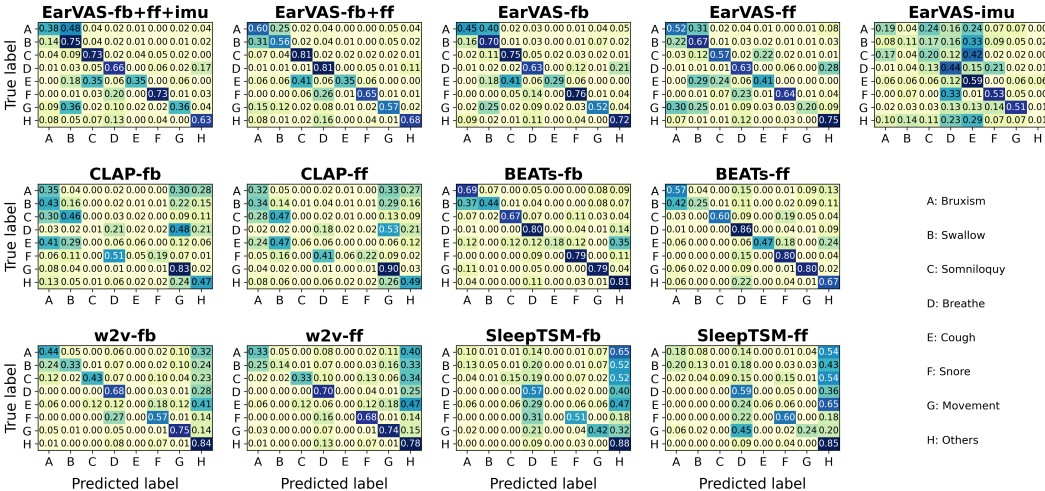

Figure 8: Confusion Matrices of the Wearer-aware Sleep Sound Event Classification Models.

**Challenges in class imbalance:** Coughing was the most underrepresented sleep event category in our dataset. Many of the models struggled to correctly identify these events, often confusing them with somniloquy or external sounds. This underscores the necessity of accounting for dataset class imbalance in wearer-aware sleep sound event classification. Although the class imbalance represents the real distribution of events in sleep scenarios, we have constructed a balanced dataset based on DreamCatcher to investigate whether models can achieve improved classification category-balanced conditions. The details are discussed in Appendix C.3.

**Challenges in jointly optimizing wearer event identification and sleep sound classification:** While traditional machine learning methods with feature engineering enable effective wearer event identification, wearer-aware sleep sound classification is a more challenging task because it also requires a model to perform sound event classification. While SleepTSM and CLAP have both been successfully used to discriminate between different sound events, introducing an additional class for background sounds and noises from the non-wearer resulted in poor accuracy. This underscores the necessity of carefully designing joint optimization methods specifically tailored for this task.

**Utility of the feedback microphone audio:** In Table 1, we found that benchmark models using only the feedback microphone audio exhibited a slightly improved performance compared to those that only used feedforward microphone audio. Upon further examination of the confusion matrices in Figure 8, we observe that utilizing feedback microphone audio can more effectively detect subtle swelling events that likely have lower intensity compared to other categories. This result highlights an important affordance of active noise-cancelling earbuds.

**Potential benefits of sensor fusion:** Comparing the performance of the EarVAS variants with different input feature modalities, we observed that leveraging both audio channels yielded higher accuracy compared to either channel alone. Integrating motion data enhanced the model's ability to discriminate wearer events from non-wearer events, which aligns with the conclusions drawn in the previous benchmark. However, the classification accuracy for categories like body movement and somniloquy decreased with this inclusion. With the proposed multi-modal DreamCatcher dataset, we

encourage researchers to explore efficient sensor fusion methods for wearer-aware sleep sound event classification.

## C.3 Wearer-Aware Sleep Sound Event Classification on Balanced Dataset

To investigate the influence on performance caused by the class imbalanced in wearer-aware sleep sound event classification, we constructed a balanced dataset based on DreamCatcher. Additionally, a comparison of the results from training Wav2Vec2.0 on both balanced and imbalanced datasets is conducted. The following describes the construction methods of the balanced dataset and notable points we found among the comparison.

**Construction of balanced training dataset based on DreamCatcher:** The categories and corresponding number of samples in the imbalanced training dataset of DreamCatcher are as follows: Bruxism 6055, Swallow 1669, Somniloquy 612, Breathe 87898, Cough 61, Snore 23626, Movement 3957. We set 10,000 as the target sample numbers for each category. For categories with more than 10,000 samples, we randomly select 10,000 samples in each epoch. We also augment the categories with fewer than this number to 10,000.

The augmentation methods conducted to the rare events are detailed as below: in terms of audio augmentation, we referred to the augmentation methods in Audiomentations [8], which include 1) gain adjustment ($\times 0.5$ to $\times 2$), 2) time shift (-0.15 to 0.15 seconds), 3) pitch shift ($\times 0.5$ to $\times 2$), 4) speed adjustment ($\times 0.5$ to $\times 2$), and 5) random masking by making 0–10% of random points zero. As for motion data, we implemented augmentation according to the methods proposed by Terry et al. [43], including jittering, scaling, magnitude-warping, time-warping, rotations among three-axis accelerometer and three-axis gyroscope respectively, and permutation.

**Evaluation of Wav2Vec2.0 on balanced and imbalanced dataset:** We trained Wav2Vec2.0 on the balanced training dataset described above with the same settings as the Wav2Vec2.0 training on the imbalanced DreamCatcher. The two models are evaluated on the same testing dataset of the raw DreamCatcher. We present a comparative results of Wav2Vec2.0 before (w2v-before-bal) and after the balancing of training dataset (w2v-after-bal) in Table 9 and Figure 9.

| Method | input channel | Evaluation Metrics (%) | | | |
|---|---|---|---|---|---|
| | | Acc. | Macro-AUC | Macro-F1 | MCC |
| w2v-before-bal | feedback | 75.45 | 91.60 | 48.36 | 56.72 |
| w2v-after-bal | feedback | 70.70 | 89.93 | 46.04 | 48.76 |

Table 9: Comparative performance of Wav2Vec2.0 training on balanced and imbalanced dataset.

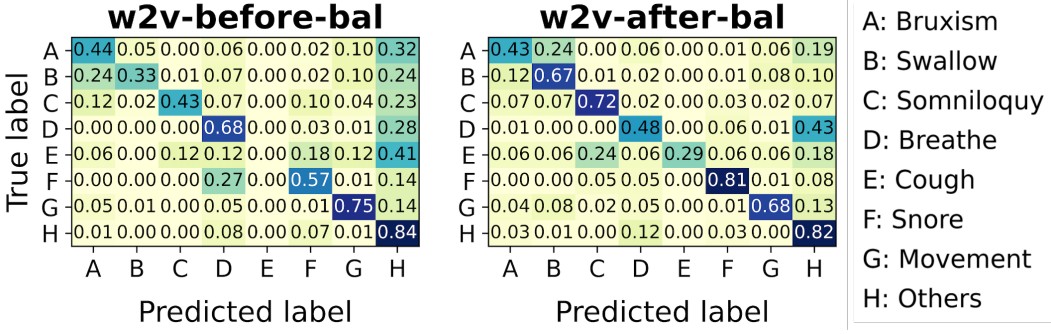

Figure 9: Confusion Matrices of Wav2Vec2.0 training on balanced and imbalanced dataset.

According to the results in the figure 9, we find that balanced dataset enhances the model's ability to learn the patterns of rare events, thereby improving the recall rate. This is particularly notable

---

[8]https://github.com/iver56/audiomentations

for the 'Cough' category. However, data balancing also led to an overall decline in performance as shown in the table 9. This also highlights the diversity of samples from downsampling categories, making DreamCatcher a valuable dataset for sleep monitoring. We envision the generation of rare sleep events, which must aligns with the patterns of human sleep, will be a highly valuable area for future research.

# D Benchmark: Wearer-Aware Sleep Sound Event Detection

## D.1 Model Architectures

Both models were built using PyTorch 1.13.0 and trained on an NVIDIA GeForce RTX 4090 GPU. We followed each paper as closely as possible and leveraged the accompanying code that was made available in both cases. Following the data format standards from the DCASE Challenges [18], we segmented the audio into 10-second clips and then generated the corresponding event onsets, offsets, and categories as labels.

**SEDNet [1]:** The original SEDNet leveraged audio sampled at 44.1 kHz, but we scaled its pre-processing steps to account for the fact that our audio was sampled at 16 kHz. Specifically, we processed audio segments by extracting log-scaled Mel-band energies using 40 bands within a 2048-point Hamming window and a 1024-point stride, yielding a $2\times155\times40$ (channel$\times$time$\times$frequency) feature vectors. We leveraged the open-sourced model found at `https://github.com/sharathadavanne/sed-crnn`, which is shared under the license found at `https://github.com/sharathadavanne/sed-crnn?tab=License-1-ov-file#readme`. The model was trained using the Adam optimizer with a batch size of 32. Training was stopped early if the Macro-F1 did not improve for 50 epochs.

**ATST-SED [38]:** For the model's CNN module, we processed audio segments by extracting 128 log-scaled Mel features from frames with a 128-ms length and a 16-ms stride. For the model's ATST-Frame module, we converted the audio segments into log-Mel spectrograms using a 64-ms Hamming window with a 10-ms stride. The resulting spectrogram comprised 64 Mel-frequency bins spanning a frequency range from 60 Hz to 7800 Hz. We leveraged the open-source model found at `https://github.com/Audio-WestlakeU/ATST-SED`, which is shared under the MIT License found at `https://github.com/Audio-WestlakeU/ATST-SED?tab=MIT-1-ov-file`. The model was trained using the Adam optimizer with a batch size of 24. We trained the first stage of ATST-SED for 200 epochs, during which the pretrained ATST-Frame module was frozen while the remaining parts were trained. Due to significant performance degradation observed during the second stage of training, we did not utilize it in this benchmark.

## D.2 Evaluation Metrics

We evaluated the models in this benchmark using collar-based metrics [21] that compare the onset and offset of a predicted and target event. An offset is often used to account for differences in prediction resolution. Based on the empirical standard proposed by Serizel et al. [37] for sound event detection, we used a 200 ms tolerance to compare onset timestamps, and we used the maximum of 200 ms and 20% of the duration of the sound event to compare offset timestamps. With these considerations in mind, we calculated the following performance metrics:

**Macro-F1.** The macro-averaged F1 score is the arithmetic mean of F1 scores in a multi-class model. In this context, true positives are defined as events in the system output that have a temporal position overlapping with the temporal position of an event with the same label in the ground truth. False positives are defined as events in the system output that have no correspondence to an event with the same label in the ground truth within the allowed tolerance, while false negatives are defined as events in the ground truth that have no correspondence to an event with the same label in the system output within the allowed tolerance. These metrics are computed on a per-class basis to produce class-specific F1 scores that are combined to calculate Macro-F1.

**Error Rate.** Error rate is calculated as the total number of substitutions, deletions, and insertions divided by the total number of events in the ground truth. Substitutions are events in the system

output with a correct temporal position but an incorrect class label, insertions are extraneous events in the system output, and deletions are events in ground truth not included in the system output.

### D.3 Supplementary Results

Table 10 examines the class-wise performance of the models that were trained in this benchmark. The rest of this subsection describes the notable trends

**Challenges in class imbalance:**    Similar to the supplementary results for the previous benchmark, class imbalance also had an impact on sound event detection. Both models performed poorly on the severely undersampled cough class, to the point that they were never able to detect such events in the dataset. However, the data imbalance issue extends beyond different categories of events. It also includes variation in the duration of individual samples and imbalances between active and inactive frames [16], which caused notable challenges for SEDNet specifically. Since our dataset reflects the natural distribution of sleep events in non-restrictive environments, we encourage the development of targeted solutions to address these challenges.

**Utility of the feedback microphone audio:**    In Table 5, we found that benchmark models using only the feedback microphone audio exhibited better performance compared to those that only used feedforward microphone audio. Upon further examination of the class-wise accuracies in Table 10, we observed that the high signal-to-noise ratio of the feedback microphone audio enables the model to detect subtle sounds. This was particularly evident with ATST-SED, which saw a marked performance boost in classes like bruxism.

**Potential benefits of sensor fusion.**    Since SEDNet supports multi-channel audio, we used it to examine the utility of dual-channel audio fusion methods. However, we found that this approach actually resulted in performance degradation. This may be because SEDNet was designed to utilize a multi-channel microphone array to localize sound sources, whereas the microphones in this application are in much closer proximity to one another and the sound source. The EarVAS model [51] for subject-aware vocal activity classification demonstrates that models intentionally trained for earbud hardware can overcome this issue. Furthermore, we did not investigate the utility of using IMU data for event detection, but given its utility in wearer event identification, it may be fruitful to pursue this direction further in a multi-modal architecture.

| Event label | F1 | Pre. | Rec. | ER. | Del. | Ins. |
|---|---|---|---|---|---|---|
| Bruxism | 12.7% | 30.4% | 8.0% | 1.10 | 0.92 | 0.18 |
| Swallow | 4.1% | 33.3% | 2.2% | 1.02 | 0.98 | 0.04 |
| Somniloquy | 5.0% | 14.3% | 3.0% | 1.15 | 0.97 | 0.18 |
| Breathe | 56.2% | 56.6% | 55.7% | 0.87 | 0.44 | 0.43 |
| Cough | 0.0% | 0.0% | 0.0% | 1.00 | 1.00 | 0.00 |
| Snore | 52.1% | 55.6% | 49.0% | 0.90 | 0.51 | 0.39 |
| Movement | 38.7% | 41.2% | 36.4% | 1.15 | 0.64 | 0.52 |

(a) ATST-SED With Feedforward Microphone Audio.

| Event label | F1 | Pre. | Rec. | ER. | Del. | Ins. |
|---|---|---|---|---|---|---|
| Bruxism | 21.6% | 30.8% | 16.7% | 1.21 | 0.83 | 0.37 |
| Swallow | 6.9% | 16.4% | 4.4% | 1.18 | 0.96 | 0.22 |
| Somniloquy | 2.8% | 20.0% | 1.5% | 1.05 | 0.98 | 0.06 |
| Breathe | 57.6% | 56.7% | 58.5% | 0.86 | 0.41 | 0.45 |
| Cough | 0.0% | 0.0% | 0.0% | 1.00 | 1.00 | 0.00 |
| Snore | 47.4% | 54.3% | 42.0% | 0.93 | 0.58 | 0.35 |
| Movement | 36.8% | 42.1% | 32.6% | 1.12 | 0.67 | 0.45 |

(b) ATST-SED With Feedback Microphone Audio.

| Event label | F1 | Pre. | Rec. | ER. | Del. | Ins. |
|---|---|---|---|---|---|---|
| Bruxism | 0.0% | 0.0% | 0.0% | 1.01 | 1.00 | 0.01 |
| Swallow | 0.0% | 0.0% | 0.0% | 1.01 | 1.00 | 0.01 |
| Somniloquy | 0.0% | 0.0% | 0.0% | 1.02 | 1.00 | 0.02 |
| Breathe | 50.8% | 50.4% | 51.2% | 0.99 | 0.49 | 0.50 |
| Cough | 0.0% | 0.0% | 0.0% | 1.11 | 1.00 | 0.11 |
| Snore | 44.2% | 48.4% | 40.6% | 1.03 | 0.59 | 0.43 |
| Movement | 3.2% | 3.3% | 3.0% | 1.86 | 0.97 | 0.89 |

(c) SEDNet With Dual-channel Audio.

| Event label | F1 | Pre. | Rec. | ER. | Del. | Ins. |
|---|---|---|---|---|---|---|
| Bruxism | 0.0% | 0.0% | 0.0% | 1.00 | 1.00 | 0.00 |
| Swallow | 0.0% | 0.0% | 0.0% | 1.00 | 1.00 | 0.00 |
| Somniloquy | 0.0% | 0.0% | 0.0% | 1.00 | 1.00 | 0.00 |
| Breathe | 49.5% | 48.9% | 50.1% | 1.02 | 0.50 | 0.52 |
| Cough | 0.0% | 0.0% | 0.0% | 1.00 | 1.00 | 0.00 |
| Snore | 44.6% | 48.1% | 41.6% | 1.03 | 0.58 | 0.45 |
| Movement | 31.7% | 31.5% | 31.9% | 1.37 | 0.68 | 0.69 |

(d) SEDNet With Feedforward Microphone Audio.

| Event label | F1 | Pre. | Rec. | ER. | Del. | Ins. |
|---|---|---|---|---|---|---|
| Bruxism | 0.0% | 0.0% | 0.0% | 1.00 | 1.00 | 0.00 |
| Swallow | 0.0% | 0.0% | 0.0% | 1.00 | 1.00 | 0.00 |
| Somniloquy | 0.0% | 0.0% | 0.0% | 1.00 | 1.00 | 0.00 |
| Breathe | 52.7% | 52.9% | 52.6% | 0.94 | 0.47 | 0.47 |
| Cough | 0.0% | 0.0% | 0.0% | 1.00 | 1.00 | 0.00 |
| Snore | 47.7% | 48.9% | 46.5% | 1.02 | 0.54 | 0.49 |
| Movement | 31.6% | 35.4% | 28.5% | 1.24 | 0.71 | 0.52 |

(e) SEDNet With Feedback Microphone Audio.

F1: F1 Score; Pre.: Precision, Rec.: Recall; ER.: Error Rate; Del.: Deletion Rate; Ins.: Insertion Rate

Table 10: Class-wise Results of the Wearer-aware Sleep Sound Event Detection Models.

