# OpenReview forum: "DreamCatcher: A Wearer-aware Multi-modal Sleep Event Dataset Based on Earables in Non-restrictive Environments"
_NeurIPS.cc/2024/Datasets_and_Benchmarks_Track — NeurIPS 2024 Track Datasets and Benchmarks Spotlight_

### Official Review · Reviewer_gmYF · 2024-06-24
**DreamCatcher: Multi-modal Sleep Dataset on Earables in Non-restrictive Environments**

**Rating:** 9
**Confidence:** 4
**Clarity:** Yes, the paper is very well-written.

**Review:**

Over the years, there are various types of technology being utilized in research and clinical studies to help detect sleep events related to sleep disorders across different age groups. In this study, the idea of using earable technology to identify specific audio and motion data is original. Moreover, the researchers used real scenarios where individuals are not sleeping alone. The researchers looked at three different sleep monitoring tasks and were able to compare and contrast across different datasets used in sleep monitoring research. It is important to address this topic because when people co-sleep, it becomes difficult to get a better understanding of one individual's sleep patterns and behaviors. However, to ask an individual to not co-sleep to measure his/her sleep is not ideal because that does not reflect their typical sleep. Measuring sleep behaviors through earable technology is a unique perspective but it may only be limited to populations such as youth and older which is mentioned in detail further below.

**Strengths:**

The following are this study's strengths:
    1) Paired or dyad dataset.
    2) Somewhat rigorous protocol where at least 6 hours of sleep in the same bedroom was required in the participants' homes.
    3) Ability to look at audio data alongside the earable so that sleep events could be heard potentially through the audio (e.g., body movements, snoring, etc.).

**Additional Feedback:**

This is a wonderful addition to the literature we have on sleep studies. The potential to use various technologies to assist in detecting sleep disorders is vast. The DreamCatcher dataset, especially with its audio data, provides a valuable lens from which we can start looking at dyadic sleep data. This can then evolve in parent-infant populations so that we can measure sleep more efficiently in the younger age groups.

**Correctness:**

I believe with the information provided, the claims made in this submission are correct.

**Documentation:**

I believe that, outside of the comments I have already made, there is enough detail on data collection and organization. The researchers have clearer made sure that participant privacy and confidentiality was maintained throughout the study and that is especially critical when doing studies in a participant's home.

**Ethics:**

I do not believe that there are any ethical concerns with the submission that warrant further discussion or review.

**Limitations:**

The authors identified some limitations of their work and offered potential solutions. However, one limitation of this work is that the presumption is for the earables to be worn and so participant compliance would be an issue. In populations where people have sensory issues and/or younger populations such as infants, the earables might need to be modified in some manner to ensure that valid and reliable data is collected. I would suggest thinking about how this technology can be used in clinical settings or leveraged in local clinics as possibly a first step to detecting a sleep disorder before a PSG is done. Access to PSGs is an issue so can the earable be used in a way so that it can help identify at-risk populations who can then be referred for a PSG?

**Opportunities For Improvement:**

While this work is another step towards evaluating sleep events, there are certain limitations. Primary among them is the lack of information provided in the protocol. How were the participants recruited? What were the results of the PSQI questionnaire? Since one of the dyads reported a sleep disorder, there is no information about the type of sleep disorder, current treatment for the disorder, etc. This background information is important especially if one was to try and reproduce the results.

**Relation To Prior Work:**

The authors have clearly discussed the context of prior work and highlighted the significance of the current study.

**Summary And Contributions:**

There are many different methods to measure sleep quality and quantity. Poor sleep can encompass a wide variety of events involving body movement, respiration, and much more. The golden standard for evaluating sleep disorders is through polysomnography (PSG) which can be done in-person at a clinic or at the patient's home (ambulatory). In this study, the authors utilize a novel concept of earbuds with sensors to detect sleep events among individuals who may sleep together. This is an important topic because in Western cultures, the supposed norm is sleeping alone and not with anyone else. However, there are situations where bed-sharing and/or room-sharing are involved (e.g., mother-child, couples, roommates, etc.) and enhanced techniques are required to properly measure sleep events. The DreamCather dataset is an attempt to look at 12 pairs of participants (N = 24 participants total) synchronized sleep data.

---

> ### Author Rebuttal · Authors · 2024-08-16
>
> We appreciate the time you took to carefully review our work. Your positive feedback contribute significantly to our team's motivation and enthusiasm for continuing our work in this field. Here is our response to your comments:
>
> **[Improvement] Background information:** We recruited participants for our study through campus user study platforms, which included students, faculty, and their family members. We did not focus on the PSQI results in our paper, as the study's primary aim was not to assess sleep quality. However, we will include this data in our dataset for the benefit of other researchers who may find it valuable. Below is the self-reported sleep disorder information alongside sleep apnea detections made by the PDX device for each participant:
>
> | Participant 1 | reported disorder | Apnea (Y/N) | Participant 2 | reported disorder |
> |---------------|-------------------|-------|---------------|-------------------|
> | 1-1             | Snore/Bruxism/Somniloquy               | Y    | 1-2            | -                |
> | 2-1             | Snore               | Y    | 2-2            | -                |
> | 3-1             | Bruxism               | N    | 3-2            | -                |
> | 4-1             | Snore/Bruxism               | Y    | 4-2            | -                |
> | 5-1             | -               | N    | 5-2            | -                |
> | 6-1             | Snore               | N    | 6-2            | -                |
> | 7-1             | Snore/Bruxism              | N    | 7-2            | Snore                |
> | 8-1             | Snore               | Y    | 8-2            | Snore                |
> | 9-1             | Bruxism/Somniloquy               | N    | 9-2            | -                |
> | 10-1             | Bruxism/Somniloquy               | N    | 10-2            | Somniloquy                |
> | 11-1             | Snore               | Y    | 11-2            | -                |
> | 12-1             | Snore/Bruxism               | Y    | 12-2            | -                |
>
> **[Limitation]:** Thank you for your insightful comments. We recognize the importance of participant compliance and the challenges presented by special populations, such as those with sensory issues and infants. Additionally, we see great potential in integrating earables into clinical settings as a preliminary screening tool for sleep disorders. This can be a promising solution for current challenges that most sleep disorders are underdiagnosed and untreated. We will actively persue collaborations with local sleep clinics to investigate this possibility further.

---

### Official Review · Reviewer_46zb · 2024-07-11
**Interesting dataset and application but subpar analysis**

**Rating:** 6
**Confidence:** 4
**Correctness:** Yes, the claims seem well-supported.
**Clarity:** The paper is written clearly enough, …

**Review:**

The application of utilizing earbuds to collect data relevant to sleep events is an interesting but under-research area, and I believe the dataset provided in here is well built for promoting the community to continue researching on this topic. The dataset is collected from 24 participants, which is not a large number, but the length of signals provided are sufficient (420 hours). The authors also conducted multiple bench-marking experiments on different tasks: wearer identification, sleep-event classification, and sleep-event detection. The performances of these models are subpar, which is understandable given the nature of the dataset.

However, I do not think this is the correct venue for this work. I would have loved to see this work be more comprehensive and flushed out, using more state-of-the-art machine learning/deep learning techniques such as transformers, transfer learning or domain adaptation approaches, to achieve great modeling performance, before being published as a dataset paper. The problem with publishing this paper as an entirely dataset paper is that it's too small and too specific for general machine learning development purposes. The number of participants is also too small to develop any generalizable model.

**Strengths:**

1. Well-curated open access dataset
2. Interesting application and potential for wide usage
3. Sufficient number of bench-marking experiments

**Additional Feedback:**

None

**Documentation:**

There is sufficient detail.

**Ethics:**

No, I do not believe there is any ethical concern.

**Limitations:**

I don't think the authors have adequate addressed the limitations of this study. The privacy concerns cannot really be addressed, as it is a common problem with all ubiquitous computing systems, so this is excusable. However, I could not find work the authors have put into addressing the dataset imbalance, which could have been accomplished through resampling methods, generative models or data augmentation methods.

**Opportunities For Improvement:**

1. I saw several small errors such as typos, wrong punctuations, potentially wrong terms used, and unexplained acronyms
2. The codebase is unorganized (please remove the .DS_Store files)
3. The codebase does not provide any code for reading the data and the experiments conducted
4. Like I said in the general review, I think the future experimentation and model development parts of this study are way more valuable than the dataset itself.

**Relation To Prior Work:**

The authors have clearly explained why their dataset is unique compared to already published datasets.

**Summary And Contributions:**

The authors collected, curated and published a dataset of in-ear wearable signals collected from 24 subjects, for the purpose of wearable sleep event detection. While the dataset is very interesting and the application intriguing, I believe the authors could have done a better job at providing more comprehensive modeling analysis.

---

> ### Author Rebuttal · Authors · 2024-08-17
>
> We sincerely appreciate your critique of our work and your thoughtful suggestions. We would like to clarify a few points that may have been misunderstood, hoping this will enable you to re-evaluate our work more accurately.
>
>
> **[W1] Incorrect venue:** In the absence of a specific list of topics for the Datasets and Benchmarks Track, we refer to the broader themes outlined by [NeurIPS 2024](https://neurips.cc/Conferences/2024/CallForPapers) to contextualize our submission. Our work aligns closely with the "*Machine learning for sciences (e.g. climate, **health**, life sciences, physics, social sciences)*" category. Sleep, occupying roughly 1/3 of human life, is critical to health. More than 1/7 of the global population suffers from at least one kind of sleep disorder yet many are undiagnosed, which makes sleep disorder a significant yet often overlooked health issue.
>
> According to [the original blog post](https://neuripsconf.medium.com/announcing-the-neurips-2021-datasets-and-benchmarks-track-644e27c1e66c) of starting this track, DreamCatcher directly responds to the call for datasets that are "*representative of real applications*" and address the issue of "*many algorithms are only evaluated on toy problems or data that is plagued with bias*". By collecting sleep data in pairs, DreamCatcher provides a more realistic representation of nocturnal environments, moving beyond the limitations of previous works that relied on performed sleep sounds or single-occupancy settings.
>
> While it's true that sensing and health topics have historically been less prominent at NeurIPS compared to fields like NLP/CV/etc., it is a non-trivial topic that has gained certain recognition from the NeurIPS community. For example, [1'] presents a multi-modal sensing dataset for stress issue, [2'] presents a tool-box for remote PPG sensing (detect blood volume sensing), and etc.
>
> **[W2] SOTA techniques:** We would like to clarify the structure of our experiments and the used techniques. DreamCatcher facilitates three distinct tasks, each with unique requirements, as outlined in Section 4. For tasks demanding advanced methodologies, such as the Classification task (Section 4.2) and the Sleep Event Detection task (Section 4.3), we have tested some state-of-the-art baselines. Importantly, for all three tasks, the ideal model should be small yet effective to deploy on resource-constrained earable system.
>
> For the Classification task, we evaluated models across a spectrum of sizes—from small to large—based on their parameter count. Notably, we have employed transformer-based pre-trained models such as BEATs[11], Wav2Vec2.0[6], and CLAP[46]. BEATs and Wav2Vec2.0 are fine-tuned with transfer learning to adapt to the sleep classification task. These models are integrated with advanced techniques, including self-distilled tokenizers, quantized latent representations,  contrastive pretraining, etc.
>
> In contrast, the Sleep Event Detection task is less developed in the literature, presenting a scarcity of benchmark models for comparison. Despite this, we have adapted ATST-SED[39], which utilizes the transformer-based architectures of BEATs[11] and ATST-Frame[3'] for frame-level feature input, demonstrating our commitment to leveraging the latest techniques.
>
> For the Wearer Event Identification task, we deliberately chose a combination of feature engineering and traditional machine learning algorithms to achieve high performance with minimal computational demand. The primary challenge here is the design of features that are computationally efficient for wearable devices with limited processing capabilities, capable of distinguishing events caused by the sleep partner.
>
> We are open to integrating additional advanced models into our benchmarks and welcome any recommendations that the reviewers may have.
>
> **[W3] Dataset size:** While the development of generalized models is valuable, the creation of specialized models for key domains is equally crucial. As illustrated in Table 1, DreamCatcher stands as the most extensive open-source sleep dataset for commercial devices, with 420 hours of dual-channel acoustic and motion data, annotated with fine-grained labels. This positions DreamCatcher favorably among well-established audio datasets: VGG-Sound (550 hours), FSD-50K (120 hours), UrbanSound8k (9 hours), ESC-50 (3 hours), etc. Meanwhile, AudioSet, which is commonly used for audio pre-training, contains approximately 5500 hours of audio data.
>
> DreamCatcher collects data from 24 participants (p=24, h=420). This number is well within the range of participant counts from previous sleep studies using commercial devices, as detailed in Table 1. Furthermore, it is comparable to other datasets at previous NeurIPS conferences that require user studies to collect data: StressID[1'] (p=65, h=39), M4Singer[4'] (p=20, h=30), Kitchens[5'] (p=16, h=8), mRI[6'] (p=20, 160k frames), etc.

---

> > ### Comment · Reviewer_46zb · 2024-08-17
> > **Great clarification**
> >
> > I apologize for misunderstanding the nature of the models used in the benchmark process and I am happy to receive a clarification of the study's analysis method. The authors have also clearly addressed the writing issues, so I am improving my overall evaluation of this paper.

---

> ### Author Rebuttal · Authors · 2024-08-17
>
> **[Improvement1] Writing errors:** Thank you for bringing these issues to our attention. We will ensure a thorough review for grammatical accuracy, correct terminology, and clarity of acronyms before the document reaches its camera-ready version.
>
> **[Improvement2 & 3] Codebase:** Thank you for your kind reminder, we have removed any extraneous files from the anonymized repository.
>
> For optimal reading and visualization of the sample data, we recommend using Audacity, which is freely available at [their official website](https://www.audacityteam.org/). As demonstrated in Figure 5, Audacity allows users to replicate our annotators' process by simply dragging audio and IMU files into the interface, displaying the audio as a spectrogram, and importing the label file via the top bar. To ensure clear audio playback, IMU channels should be muted. We have decided against open-sourcing the visualization code that produces spectrograms akin to those in Figures 2 and 6. This decision is based on the efficiency of Audacity for such tasks, which is significantly faster than running the equivalent Python script.
>
> The complete experiment code, which enables the reproduction of our paper's results, will be made publicly available with the full dataset upon the final paper submission. We refrained from including this code in the anonymized repository to avoid confusion, as it is not possible to train a model with this small subset of data.
>
> **[L1] Privacy concern:** We agree that the privacy issues associated with wearable devices fall outside the scope of our dataset provision work. However, we assure that all data released has been processed properly to uphold stringent privacy protection standards.
>
> **[L2] Dataset imbalance:**
>
> DreamCatcher is curated to authentically represent the natural scarcity of certain sleep events, which do not occur in isolation but rather unfold throughout a night's sleep. This natural temporal distribution is crucial; thus, having a dataset with only short event windows without the context of what happened before and after does not make sense. To put it another way, if a given person has a rare event, then producing a balanced dataset from their data would require getting a small handful of non-rare events. leading to very little data from that person in general. By balancing the dataset according to events, you would now have a dataset that is imbalance according to participants.
>
> In light of these considerations, we trained our benchmark models solely on the unaltered DreamCatcher dataset, without resorting to data augmentation. While data augmentation could potentially enhance performance on infrequent events, our primary objective is to present a sleep dataset that mirrors the real-world rarity of certain sleep conditions.
>
> We acknowledge the potential of data augmentation in improving classification tasks and recognize that our community may further this research. To contribute to this dialogue, we will conduct a baseline experiment by fine-tuning Wav2Vec2.0 with data augmentation during the rebuttal period. The outcomes and details of this experiment will be shared below upon completion of the training and evaluation phases.
>
>
>
> [1'] Chaptoukaev, H., Strizhkova, V., Panariello, M., Dalpaos, B., Reka, A., Manera, V., ... & M Ferrari, L. (2024). Stressid: a multimodal dataset for stress identification. Advances in Neural Information Processing Systems, 36.
>
> [2'] Liu, X., Narayanswamy, G., Paruchuri, A., Zhang, X., Tang, J., Zhang, Y., ... & McDuff, D. (2024). rppg-toolbox: Deep remote ppg toolbox. Advances in Neural Information Processing Systems, 36.
>
> [3'] Shao, N., Li, X., & Li, X. (2024, April). Fine-tune the pretrained atst model for sound event detection. In ICASSP 2024-2024 IEEE International Conference on Acoustics, Speech and Signal Processing (ICASSP) (pp. 911-915). IEEE.
>
> [4'] Zhang, L., Li, R., Wang, S., Deng, L., Liu, J., Ren, Y., ... & Zhao, Z. (2022). M4singer: A multi-style, multi-singer and musical score provided mandarin singing corpus. Advances in Neural Information Processing Systems, 35, 6914-6926.
>
> [5'] Tanke, J., Kwon, O. H., Mueller, F. B., Doering, A., & Gall, J. (2023). Humans in kitchens: a dataset for multi-person human motion forecasting with scene context. Advances in Neural Information Processing Systems, 36, 10184-10196.
>
> [6'] An, S., Li, Y., & Ogras, U. (2022). mri: Multi-modal 3d human pose estimation dataset using mmwave, rgb-d, and inertial sensors. Advances in Neural Information Processing Systems, 35, 27414-27426.

---

> > ### Author Rebuttal · Authors · 2024-08-17
> >
> > Here is the method we used for data augmentation: In terms of audio augmentation, we primarily referred to the augmentation methods in [Audiomentations](https://github.com/iver56/audiomentations), which include 1) gain adjustment (x0.5 to x2), 2) time shift (− 0.15 to 0.15 seconds), 3) pitch shift (x0.5 to x2), 4) speed adjustment (x0.5 to x2), and 5) random masking by making 0–10% of random points zero. As for motion data, we implemented augmentation according to the methods proposed by Terry [7'] et al., including jittering, scaling, magnitude-warping, time-warping, rotations among three-axis accelerometer and three-axis gyroscope respectively, and permutation.
> >
> > [7'] Terry T. Um, Franz M. J. Pfister, Daniel Pichler, Satoshi Endo, Muriel Lang, Sandra Hirche, Urban Fietzek, and Dana Kulić. 2017. Data augmentation of wearable sensor data for parkinson’s disease monitoring using convolutional neural networks. In Proceedings of the 19th ACM International Conference on Multimodal Interaction (ICMI '17). Association for Computing Machinery, New York, NY, USA, 216–220. https://doi.org/10.1145/3136755.3136817

---

> > ### Comment · Reviewer_46zb · 2024-08-17
> > **Additional analysis added**
> >
> > With the additionally analysis provided I am happy to improve my overall evaluation.

---

> > > ### Author Rebuttal · Authors · 2024-08-30
> > >
> > > We sincerely apologize for the delayed rebuttal due to issues with our computational resources. With your assistance, we have recognized that techniques for balancing datasets can provide valuable references for future researchers developing methods on the DreamCatcher dataset. We will incorporate this point into our revised paper. We have constructed a balanced training dataset following the method demonstrated in the previous rebuttal, and the experimental results along with our analysis are attached in the PDF file.

---

### Official Review · Reviewer_VByL · 2024-07-21
**This paper introduces DreamCatcher, the first open dataset for wearable ear-worn device-based sleep event detection. It addresses limitations of existing algorithms by including real-world data with interference from non-wearers. The dataset, featuring 420 hours of audio and motion data, advances wearer-aware sleep event monitoring. Given its contributions and minor limitations, I rate this paper a 7.**

**Rating:** 7
**Confidence:** 4
**Correctness:** Constructed in a reasonable manner

**Review:**

The authors described the datasets used in previous sleep monitoring research and the multiple challenges they faced, such as reliance on specialized equipment and ignoring interference caused by non-wearers. The authors introduced the first open dataset developed for wearable ear-worn device-based sleep event detection algorithms, addressing this gap. The finely annotated labels of eight different sleep events contribute to advancing research on wearer-aware sleep event monitoring.

**Strengths:**

1. The data is rich and diverse. The DreamCatcher dataset includes extensive synchronized two-channel audio and motion data, covering various types of sleep events. It reflects the natural scarcity of certain sleep disorders, such as bruxism, swallowing, and coughing.

2. The DreamCatcher dataset includes data collected from 12 pairs of participants, totaling 420 hours, making it the largest sleep dataset to date. More importantly, it consists of real-world data and is fully open-source.

3. The binary classification task focuses on determining whether audio events originate from the wearer or other sources, an aspect not previously explored in other studies.

**Additional Feedback:**

No additional feedback.

**Clarity:**

The paper is clear and explains the motivation for constructing the dataset and the process of constructing it.

**Documentation:**

The authors provide detailed information on the dataset, including the collection process.

**Ethics:**

No ethic problem.

**Limitations:**

Here are my questions for the authors:

1. What is the vertical axis in Figure 3(a)? How should the significance of Figure 3(a) be understood?
2. In the task of "Wearer-Aware Sleep Sound Event Detection," the existing methods ATST-SED and SEDNet were only validated on the DreamCatcher dataset. This validation alone does not fully demonstrate how DreamCatcher provides a superior evaluation environment for deep learning models compared to other existing sleep datasets, nor does it establish its efficacy as a benchmark. Could you please provide further explanations?

**Opportunities For Improvement:**

As the authors mentioned, rare events are often more significant. However, the DreamCatcher dataset is highly imbalanced (for example, coughing events occupy only 0.04 hours in total). It is necessary to curate a balanced supplementary dataset that provides more comprehensive information on various events.

**Relation To Prior Work:**

The author makes a thorough comparison with previous work and explains the differences from previous work.

**Summary And Contributions:**

Sleep monitoring is crucial for facilitating diagnosis, and lightweight ear-worn devices can provide convenient real-time human activity monitoring. Existing acoustic-based sleep event detection algorithms mainly focus on audio feature engineering and lightweight deep learning models, which are typically developed using data collected in controlled environments with minimal interference. However, in reality, people often share sleeping spaces with roommates or partners, whose movements and sounds can interfere with the monitoring results of the wearer. To address this issue, the authors introduced DreamCatcher, the first open dataset specifically developed for wearable ear-worn device-based sleep event detection. This dataset aims to inspire researchers to further explore efficient human voice activity detection using ear-worn devices.

---

> ### Author Rebuttal · Authors · 2024-08-17
>
> Dear reviewr VByL, we sincerely thank you for your support of our work and appreciate your thoughtful suggestions that have helped us improve. Here are our responses to the points you raised:
>
> **[Improvement] Dataset imbalance:** DreamCatcher is collected to accurately represent the inherent rarity of specific sleep events which occur as part of a complete night's sleep cycle. These events are distributed naturally across the time axis, and it is essential that our dataset maintains this context. A dataset limited to isolated short event windows would lack the necessary context of preceding and succeeding events, thereby distorting the natural event sequence.
>
> Furthermore, the creation of a balanced dataset for infrequent events through artificial means such as synthetic data generation or human intervention constitutes a separate research endeavor, one that holds its own merit and complexity.
>
>
> **[L1] Figure interpretation:** Thank you for highlighting the issue with Figure 3(a). The vertical axis should be properly displaying "duration (s)". We will fix this issue in the camera ready version.
>
> Both Figures 3(a) and 3(b) are designed to showcase the data distribution in a manner that reflects the inherent characteristics of sleep events. Specifically, Figure 3(b) illustrates the overnight distribution of seven types of sleep events. This distribution corroborates well-established sleep patterns. For instance, movements and swallowing are more prevalent at the onset of sleep as individuals settle in, snoring and audible breathing typically occur after the sleeper has fallen asleep, and somniloquy often emerges during the deep sleep phase in the latter part of the night.
>
> **[L2] SED experiment:** To the best of our knowledge, as shown in Table 1 in the paper, our proposed DreamCatcher is the first publicly available sleep dataset that includes the audio modality. Therefore, in our defined "Wearer-Aware Sleep Sound Event Detection" task, we did not evaluate the benchmark models on other sleep datasets. However, our benchmark models have indeed performed excellently in the sound event detection task. The ATST-SED model, which is based on the Transformer architecture and utilizes transfer learning techniques, ranks #1 in the DCASE2023 Challenge Task 4 (https://paperswithcode.com/sota/sound-event-detection-on-desed).
>
> On the other hand, the CRNN architecture of SEDNet has been proven to be highly effective in sound event detection. We strictly followed the settings and replemented benchmark models on the DreamCatcher dataset, yet we obtained results that appear to be less than satisfactory. This indicates that the "Wearer-Aware Sleep Sound Event Detection" task requires more specific and effective methods to be addressed. Given the importance of this task as discussed in our paper, we define this task and implement benchmark models in hopes of encouraging more researchers to focus on this task and propose more effective methods.

---

> > ### Comment · Reviewer_VByL · 2024-08-25
> >
> > The author has adequately addressed the issues I raised, so I confirm my original score.

---

### Official Review · Reviewer_ZSYL · 2024-07-25
**A good dataset but the open research issues that can be explored have not been discussed**

**Rating:** 7
**Confidence:** 5

**Review:**

The proposed dataset is interesting in the sense that it captures eight different sleep disorders, and the authors have used a setup with multimodality and multiuser to collect large-scale sensing data. The authors have also demonstrated the annotation procedure used, and then evaluated the dataset with some available benchmark models. Overall, the paper has been presented nicely, and the authors did a good job in explaining the data collection and annotation procedure.

However, I feel that there are a few shortcomings which are there in the current version of the paper.

1. Some details are missing; for example, what is the percentage of disagreement among the annotators? Have the authors did any sanity check to verify that the majority voting works when there is an disagreement? If so, how?

2. What is the broad utility of this dataset? I understand that the dataset can be used for sleep disorder classification. However, this problem has been well-studied in the literature and there are multiple models for it. Can there be any new research directions that can be explored with this dataset? How does the ML community benefit from the publication of this dataset?

3. The documentation of the dataset is poor. Neither the paper nor the dataset link explicitly states what fields are there in the dataset, how the missing values (if any) are handled, how the dataset is organized, etc.

4. How do the authors ensure time synchronization across multiple modalities, which can be a major issue with multimodal data collection through multiple independent devices?

**Strengths:**

+ A multi-modal multi-user dataset for sleep disorder classification
+ Overall, the paper is well-written (although some explanations are missing)

**Additional Feedback:**

Please check my detailed comments above.

**Clarity:**

The descriptions given in the paper are mostly clear, although explanations are needed in some parts (check my detailed review).

**Correctness:**

It is not clear how the authors have ensured synchronization across multiple modalities. The authors mentioned that the data has been synchronized and the participants had set an alarm, but further details are missing. For example, how has the clock drift across hardware been handled? Such issues might question the overall correctness of the collected data.

**Documentation:**

Documentation is not at all good. The different fields in the data files are not clear.

**Ethics:**

Ethical issues have been addressed in the paper.

**Limitations:**

1. Some details are missing; for example, what is the percentage of disagreement among the annotators? Have the authors did any sanity check to verify that the majority voting works when there is an disagreement? If so, how?

2. What is the broad utility of this dataset? I understand that the dataset can be used for sleep disorder classification. However, this problem has been well-studied in the literature and there are multiple models for it. Can there be any new research directions that can be explored with this dataset? How does the ML community benefit from the publication of this dataset?

3. The documentation of the dataset is poor. Neither the paper nor the dataset link explicitly states what fields are there in the dataset, how the missing values (if any) are handled, how the dataset is organized, etc.

4. How do the authors ensure time synchronization across multiple modalities, which can be a major issue with multimodal data collection through multiple independent devices?

**Opportunities For Improvement:**

I have a feeling that the overall scope of the paper is limited. The paper proposed a dataset for sleep disorder classification, but beyond that, it is not very clear about what can be the other downstream tasks that can be explored with this dataset. More importantly, it is not clear how the dataset can benefit the ML and the pervasive/ubiquitous computing research community. It would be good if the authors can provide such details in the discussion.

Also, it would be useful to provide several missing details, as I have pointed out in my detailed review.

**Relation To Prior Work:**

Related works have been explained properly.

**Summary And Contributions:**

This paper discusses a multi-modal multiuser dataset for sleep disorder detection, where the authors have used a earable device to collect the dataset. The authors have also reported the data annotation procedure adopted, and discussed performance benchmarks with various other models.

---

> ### Author Rebuttal · Authors · 2024-08-17
>
> We appreciate your questions and comments, providing us with an opportunity to further elucidate specific facets of our research. Here is our detailed response to the points you have raised:
>
> **[L1] Annotation details:** Our hierarchical inspection process for annotations consists of three steps. Initially, raw annotations are provided by first-tier annotators. These are then reviewed and refined by experienced annotators. Finally, a rigorous check is conducted by three of our authors, ensuring each annotation is reviewed by at least one author. Throughout this process, every label undergoes a minimum of three checks.
>
> In the early stages of developing our annotation pipeline, any discrepancies noted by an author against the second step's annotations were resolved through discussion and voting among the authors. As the process matured, the authors began to resolve similar issues by referring to the outcomes of past votes, while continuing to discuss and vote on new discrepancies that arose.
>
> We do not have records of the revisions made by the second step's annotators, and therefore cannot directly report a 'percentage of disagreement' among all annotators. However, we can offer the revision rate from the final check as an approximate measure of disagreement. This rate was calculated by comparing 21,953 sleep events in 28 hours of 2nd step annotations with the final annotations, which revealed a revision rate of 9%.
>
>
> **[L2] Dataset contribution:** We would like to highlight that prior work has always assumed that there is a single person in the room, which is limited and impractical for everyday situations. As reviewer gmYF summarized: "*It is important to address this topic because when people co-sleep, it becomes difficult to get a better understanding of one individual's sleep patterns and behaviors. However, to ask an individual to not co-sleep to measure his/her sleep is not ideal because that does not reflect their typical sleep.*" Such limitations are compounded by the lack of reproducibility as both datasets and models were proprietary in prior work. Our study addresses these shortcomings by introducing a sleep dataset featuring paired participants and fine-grained event annotations.
>
> We recognize the value of generalized datasets to the machine learning community, but we also stress the importance of specialized datasets for critical areas such as sleep research. Sleep takes up approximately 1/3 time of a human's entire life. More than one-seventh of the global population suffers from at least one kind of sleep disorder, yet most are undiagnosed due to underestimation and prohibitive costs of PSG sessions. Wearable devices, like earbuds, provide a cost-effective preliminary screening tool for sleep disorders.
>
> DreamCatcher is assessed across three main tasks: wearer-awareness, sleep event classification, and sleep event detection. (1) wearer-awareness is crucial for ensuring that acoustic-based functions on wearable devices respond exclusively to the wearer's activity. (2) by including audio and motion data for sleep event classification, DreamCatcher not only serves its primary purpose but can also be used as a domain-specific test set for general audio models or multi-modality sensing models, e.g. the zero-shot evaluation of CLAP in Table 4. (3) as an underrepresented area in research, sound event detection can benefits from DreamCatcher's comprehensive 420 hours of dual-channel audio data. The inclusion of motion data further promotes the exploration of modality fusion in event detection tasks.
>
>
> **[L3] Dataset documentation:** For the documentation of the dataset, we followed the submission guidelines to provide a croissant metadata in the supplementary material. We now uploaded a duplicate to the dataset link.
>
> **[L4] Modality synchronization:** We acknowledge the need for clarity in our explanation of the data alignment process (lines 139-145) and provide the following refined description:
>
> - Alignment type 1: aligning data modalities in each earbud.
>     - Necessity for alignment: Although data is collected in pairs, algorithms will only take data from one target participant at a time. The audio and imu data should be intrinsically synchronized because they are connected to the same ESP32 board and controlled by the same microcontroller.
>     - Potential misalignment: Inherent clock drift between the audio and IMU sampling protocols is observed, with the IMU recording potentially extending 3 seconds longer over 7 hours compared to the audio data, a deviation of about 0.01%.
>     - Solution: Because this drift is evenly distributed over the entire recording, we correct it by re-scaling the IMU data to match the audio recording duration.
> - Alignment type 2: aligning each pairs' data.
>     - Necessity for alignment: Aligning the data from both participants in a pair is essential for the annotators to determine the source of the events, as exampled in Appendix B.2.2. This facilitates a coherent annotation interface, as shown in Figure 5, which requires less alignment granularity than Type 1.
>     - Potential misalignment: Each participant's data is independently recorded by separate earbuds, often started at different times.
>     - Solution: Without a simultaneous start, the aforementioned alarm clock can be a compensatory reference. We manually adjusted the audio recordings to align the alarm clock's spectrogram and ensure simultaneous playback without perceptible echo, confirming minimal time discrepancy.

---

> > ### Comment · Reviewer_ZSYL · 2024-08-19
> >
> > Thanks for submitting the rebuttal and the detailed response. I agree to most of your points. However, while reading your rebuttal, particularly, the dataset contributions (which is my main concern), I got a few followup thoughts.
> >
> > 1. I agree to you that analyzing co-sleep events for multi-user scenarios is important; however, is earable a very suitable device for the same? Will not the people feel uncomfortable while sleeping with a pair of earables, and will not the device itself affect their sleep states? There have been a lot of works on non-intrusive or passive monitoring of sleep events, through WiFi sensing or mmWave sensing (check https://doi.org/10.1145/3491245), even for multi-user scenarios. A major criticism for wearable technologies for sleep event detection is that they create discomforts, primarily during the sleep, and I feel that earable will not be free from that, or may even worsen the scenario compared to other wearables like Smartband or Smart ring. I would request the authors to comment on the same. Note that I agree that earable might be able to provide richer insights compared to smartwatch, smart ring, or even the passive sensing modalities, as they are closer to eye, ear and the head -- however; I am more concerned about their practical usability and consequently the usefulness of the dataset. While clarifying this, it would also be good if the authors can comment on how they have ensured that the data collection procedure itself didn't affect the sleep events.
> >
> > 2. While I am fine with a focused dataset (in this case, sleep event monitoring), I was thinking whether the dataset can be used to broaden the possible areas that can be explored. Note that you have collected dataset from earables; so with proper annotations, the dataset might also be explored for other research areas, possibly like heart-related events or digestion-related events, which might have an impact on the ear canals. This way, you'll be able to convince the readers more about the usability of the dataset.
> >
> > Sorry for putting up these additional queries, but I feel they are important from the perspective of the paper's completeness and usefulness for the broad audience. I'll be happy to increase my rating on this paper, but I would really like to see the authors' comments on these issues.

---

> > ### Author Rebuttal · Authors · 2024-08-24
> >
> > Dear reviewer ZSYL, here is our response to your questions:
> >
> > **[Q1]** Thanks for pointing out the variety of approaches to sleep monitoring, with contactless methods ranking high in comfort. To address the question, we will first explore these contactless techniques before elucidating the comfort provided by sleep earbuds.
> >
> > **Contactless sleep monitoring and multi-user case:** Contactless sleep monitoring offers two primary methodologies: acoustic-based and reflection-based. Acoustic-based monitoring is prone to interference in multi-user settings. In contrast, reflection-based methods, which emit waves and analyze the returned signals to detect motion, are capable of tracking movements, as well as respiration and heart rates through minor chest movements at specific frequencies. Commonly used signal includes WiFi[1'], mmWave[2'], sonar system[3'], etc. Notably, with locating the monitoring subject in advance, these methods, particularly those referenced in [2'] and [3'], can manage multi-user environments provided there is a minimum 20cm distance between individuals, as mentioned in [3']. However, these technologies are less effective for detecting acoustic sleep events such as snoring, swallowing, or somniloquy, which are vital for diagnosing sleep disorders.
> >
> > In summary, both earbuds and reflection-based contactless methods show promise in addressing the multi-user challenge, yet cater to different aspects of sleep monitoring. Moreover, as indicated in [4'], there is an inherent trade-off between accuracy and comfort.
> >
> > **Sleep earbuds' comfort level:** The prototype earbuds used in our study, as well as everyday commercial Bluetooth earbuds, may be less comfortable for sleep, particularly for those who struggle to fall asleep. Nevertheless, there are specifically designed, market-available options that offer greater comfort for sleeping, such as [Amazfit Zenbuds](https://www.amazon.com/Amazfit-Blocking-Soothing-Comfortable-Detection/dp/B08H555KLK) and [Bose Sleepbuds](https://www.amazon.com/Bose-Sleepbuds-II-technology-Clinically/dp/B08FRR6Z1N?th=1). These earbuds are small, soft, and ergonomically designed to enhance comfort during sleep, indicating a promising future for sleep monitoring via earbuds.
> >
> > In our study, we informed potential participants about the possible discomfort associated with wearing the earbud and the PDX device during sleep. Consequently, those who chose to participate were less likely to be affected by such discomfort. Post-experiment feedback on comfort levels revealed that the participant using only the earbud experienced slight discomfort, whereas the participant wearing both the PDX device and the earbud reported more discomfort, primarily due to the PDX device. This discomfort aligns with the 'first-night effect,' an inevitable issue in professional sleep monitoring devices, as noted in [30]. Despite this, the data collected was sufficient and matched the self-reported sleep patterns of the participants.
> >
> > **[Q2]** While we maintain that detecting sleep disorder events is important, we agree with your point that a dataset with wider applications would be advantageous. However, since DreamCatcher is already annotated, it is not feasible to gather new channel data to serve as ground truth.
> >
> > One potential method for DreamCatcher to offer additional annotations involves the participant who wore the PDX device[1]. This device can provide ground truth data for respiratory rate (RR), heart rate (HR), airflow, blood oxygen, a rough estimate of sleep stages, and an additional input signal for SpO2. However, due to the lack of audio recording on the PDX device, aligning its data with our earbud data may require additional effort. We are willing to open-source these data as supplementary for DreamCatcher if it is beneficial to the community.
> >
> > [1'] Zhang, F., Wu, C., Wang, B., Wu, M., Bugos, D., Zhang, H., & Liu, K. R. (2019). SMARS: Sleep monitoring via ambient radio signals. IEEE Transactions on Mobile Computing, 20(1), 217-231.
> >
> > [2'] Yang, Z., Pathak, P. H., Zeng, Y., Liran, X., & Mohapatra, P. (2017). Vital sign and sleep monitoring using millimeter wave. ACM Transactions on Sensor Networks (TOSN), 13(2), 1-32.
> >
> > [3'] Nandakumar, R., Gollakota, S., & Watson, N. (2015, May). Contactless sleep apnea detection on smartphones. In Proceedings of the 13th annual international conference on mobile systems, applications, and services (pp. 45-57).
> >
> > [4'] Hussain, Z., Sheng, Q. Z., Zhang, W. E., Ortiz, J., & Pouriyeh, S. (2022). Non-invasive techniques for monitoring different aspects of sleep: A comprehensive review. ACM Transactions on Computing for Healthcare (HEALTH), 3(2), 1-26.

---

> > > ### Comment · Reviewer_ZSYL · 2024-08-27
> > >
> > > Thanks for the clarifications. Please include these details in your revised paper.

---

### Decision · Program_Chairs · 2024-09-26

**Decision:**

Accept (Spotlight)

**Comment:**

This paper presents a novel multimodal dataset that include, for the first time, acoustic and motion sensor data from earables (earbuds) worn by participants during their sleep.
The strengths of this paper includes:
* The dataset is quite novel and rich, provides new multimodal sensors in a relatively new and unexplored areas (using earables rather than PSG).
* The data collection set up is solid, with the inclusion of paired dyad datasets
* The tasks and benchmark are quite comprehensive. The dataset is tested against several benchmarks on three sleep event recognition tasks.
* The evaluation also includes aspects of 'wearer awareness' whereby they can distinguish ambient or non-wearer sound.
* The authors were very responsive during the rebuttal and discussion period.

Weaknesses include:
* The writeup is solid, but can be improved by providing more details on the participant background statistics, and details of their disorders (as asked by the last reviewer), details on data preprocessing, including alignment of the signals.
* Adding details on the limitations and opportunities to improve the dataset, such as more balanced dataset on the participants (with or without specific disorders or conditions).

In summary, it is a very good paper with original and significant contributions to the research community, and strong pathways to impact.
I strongly encourage the authors to improve the camera ready version by incorporating the reviewers' feedback.